# The pseudo GTPase CENP-M drives human kinetochore assembly

**Federica Basilico[1,2†], Stefano Maffini[1†], John R Weir[1†], Daniel Prumbaum[3], Ana M Rojas[4], Tomasz Zimniak[5], Anna De Antoni[2], Sadasivam Jeganathan[1], Beate Voss[1], Suzan van Gerwen[1], Veronica Krenn[1,2], Lucia Massimiliano[2], Alfonso Valencia[6], Ingrid R Vetter[1], Franz Herzog[5], Stefan Raunser[3], Sebastiano Pasqualato[2], Andrea Musacchio[1,7]\***

[1]Department of Mechanistic Cell Biology, Max Planck Institute of Molecular Physiology, Dortmund, Germany; [2]Department of Experimental Oncology, European Institute of Oncology, Milan, Italy; [3]Department of Physical Biochemistry, Max Planck Institute of Molecular Physiology, Dortmund, Germany; [4]Computational Biology and Bioinformatics Group, Institute of Biomedicine of Seville, Campus Hospital Universitario Virgen del Rocio, Seville, Spain; [5]Department of Biochemistry and Gene Center, Ludwig-Maximilians-Universität, München, Munich, Germany; [6]Structural Biology and Biocomputing Programme, Spanish National Cancer Centre–CNIO, Madrid, Spain; [7]Centre for Medical Biotechnology, Faculty of Biology, University of Duisburg-Essen, Essen, Germany

**\*For correspondence:** andrea.
musacchio@mpi-dortmund.mpg.de

[†]These authors contributed
equally to this work

**Competing interests:** The
authors declare that no
competing interests exist.

**Reviewing editor**: John Kuriyan,
Howard Hughes Medical
Institute, University of California,
Berkeley, United States

**Abstract** Kinetochores, multi-subunit complexes that assemble at the interface with centromeres, bind spindle microtubules to ensure faithful delivery of chromosomes during cell division. The configuration and function of the kinetochore–centromere interface is poorly understood. We report that a protein at this interface, CENP-M, is structurally and evolutionarily related to small GTPases but is incapable of GTP-binding and conformational switching. We show that CENP-M is crucially required for the assembly and stability of a tetramer also comprising CENP-I, CENP-H, and CENP-K, the HIKM complex, which we extensively characterize through a combination of structural, biochemical, and cell biological approaches. A point mutant affecting the CENP-M/CENP-I interaction hampers kinetochore assembly and chromosome alignment and prevents kinetochore recruitment of the CENP-T/W complex, questioning a role of CENP-T/W as founder of an independent axis of kinetochore assembly. Our studies identify a single pathway having CENP-C as founder, and CENP-H/I/K/M and CENP-T/W as CENP-C-dependent followers.

## Introduction

Mitosis creates two genetically identical daughter cells through the equal segregation of sister chromatids. Meiosis, on the other hand, aims to halve the genetic content of a mother cell. In either case, chromosome segregation is executed by molecular machinery that is largely conserved in the evolution of eukaryotes. Kinetochores are essential elements of such machinery (*Santaguida and Musacchio, 2009*; *Westermann and Schleiffer, 2013*). They are biochemically complex structures, which contain multiple copies of approximately 30 core subunits, in turn contributing to recruit many additional regulatory proteins (*Figure 1A*). Schematically, kinetochores can be viewed as layered structures, whose inner and outer layers directly contact centromeric chromatin and spindle microtubules, respectively.

The KMN network is the main constituent of the outer kinetochore (*Figure 1A*). It is a 10-subunit assembly originating from the interaction of three sub-complexes, named the Knl1-complex, the Mis12-complex, and the Ndc80-complex (*Cheeseman and Desai, 2008*). It provides at least two sites

**eLife digest** When a human cell divides to make new cells, its 46 chromosomes must be replicated and then separated evenly between the two daughter cells. The process of separation is performed by the spindle—a network of fibres that form inside the cell, attach to the chromosomes and pull the copies to the opposite ends of the cell.

The spindle fibres attach to a structure called a kinetochore, which forms at a region of the chromosomes called the centromere. The kinetochore has a layered structure with multiple copies of many proteins, and the inner layer is composed of at least 16 centromeric proteins. These proteins interact directly with the centromere and influence the formation of the rest of the kinetochore and the spindle fibres. While some of the interactions between centromeric proteins have been uncovered, the roles of several of them—including one called CENP-M—remain unknown.

Now, Basilico, Maffini, Weir et al. reveal that CENP-M is essential for assembling and stabilizing the inner layer of the kinetochore. However, while it is structurally and evolutionarily related to enzymes called GTPases, CENP-M is not an enzyme. Instead, the CENP-M protein interacts with three other centromeric proteins to form a complex that becomes part of the inner layer of the kinetochore. Basilico, Maffini, Weir et al. also find that another centromeric protein, CENP-C, appears to start the assembly of the inner layer. This protein then recruits two complexes made of other centromeric proteins to the kinetochore, including the complex that contains CENP-M.

The next challenge will be to reconstitute larger protein complexes that contain more proteins from the inner layer of the kinetochore, so that these assemblies can be studied in greater detail. It will also be important to investigate how CENP-C acts as a scaffold to organize the interface between the kinetochore and the centromere.

for microtubule attachment, positioned in the Ndc80 and Knl1 subunits, and it regulates cell cycle progression through the recruitment, activation, and subsequent silencing of the spindle assembly checkpoint (SAC) (*Foley and Kapoor, 2013*; *Figure 1A*).

The inner kinetochore controls outer kinetochore assembly (*Liu et al., 2006*; *McClelland et al., 2007*; *Cheeseman et al., 2008*), influences microtubule binding (*McClelland et al., 2007*; *Amaro et al., 2010*), and contributes to epigenetic specification of centromeres (*Takahashi et al., 2000*; *Okada et al., 2006, 2009*; *Hori et al., 2013*). The inner kinetochore is built on centromeres, genetic loci whose hallmark is the presence of CENP-A, a variant of histone H3. CENP-A interacts with histone H2A, H2B and H4 in a nucleosome-like structure whose precise organization is under discussion (*Black and Cleveland, 2011*; *Figure 1A*).

In vertebrates, a group of at least 16 inner kinetochore proteins, collectively identified as constitutive centromere-associated network (CCAN), neighbors the CENP-A nucleosome (*Obuse et al., 2004*; *Liu et al., 2005*; *Foltz et al., 2006*; *Izuta et al., 2006*; *Okada et al., 2006*). The CCAN (whose subunits are indicated as 'CENP'—for centromeric protein—followed by a letter, see *Figure 1A*) interacts with the outer kinetochore and contributes to maintain centromere identity by participating in CENP-A loading at every new cell division cycle (*Perpelescu and Fukagawa, 2011*; *Westhorpe and Straight, 2013*). Understanding the physical organization of the CCAN is instrumental to shed light on these vital functions of CCAN.

Initial characterization of CCAN subunits after enrichment from cell lysates suggested that they engage in a complex network of interactions (*Cheeseman et al., 2002*; *De Wulf, 2003*; *Obuse et al., 2004*; *Liu et al., 2005*; *Foltz et al., 2006*; *Izuta et al., 2006*; *Okada et al., 2006*). Subsequent work led to biochemical reconstitution of at least three CCAN sub-complexes (*Figure 1A*). First, the 4-subunit CENP-O/P/Q/U complex (also known as COMA complex in *S. cerevisiae*) has been implicated in microtubule binding and spindle checkpoint control (*Perpelescu and Fukagawa, 2011*), and is believed to interact with CENP-R, a protein of unknown function (*Hori et al., 2008b*). Second, the 2-subunit CENP-N/L complex binds directly to the CATD box, a specific segment of CENP-A that has been implicated in the epigenetic specification of centromeres (*Carroll et al., 2009*; *Black and Cleveland, 2011*; *Hinshaw and Harrison, 2013*). Third, the 4-subunit CENP-T/W/S/X complex

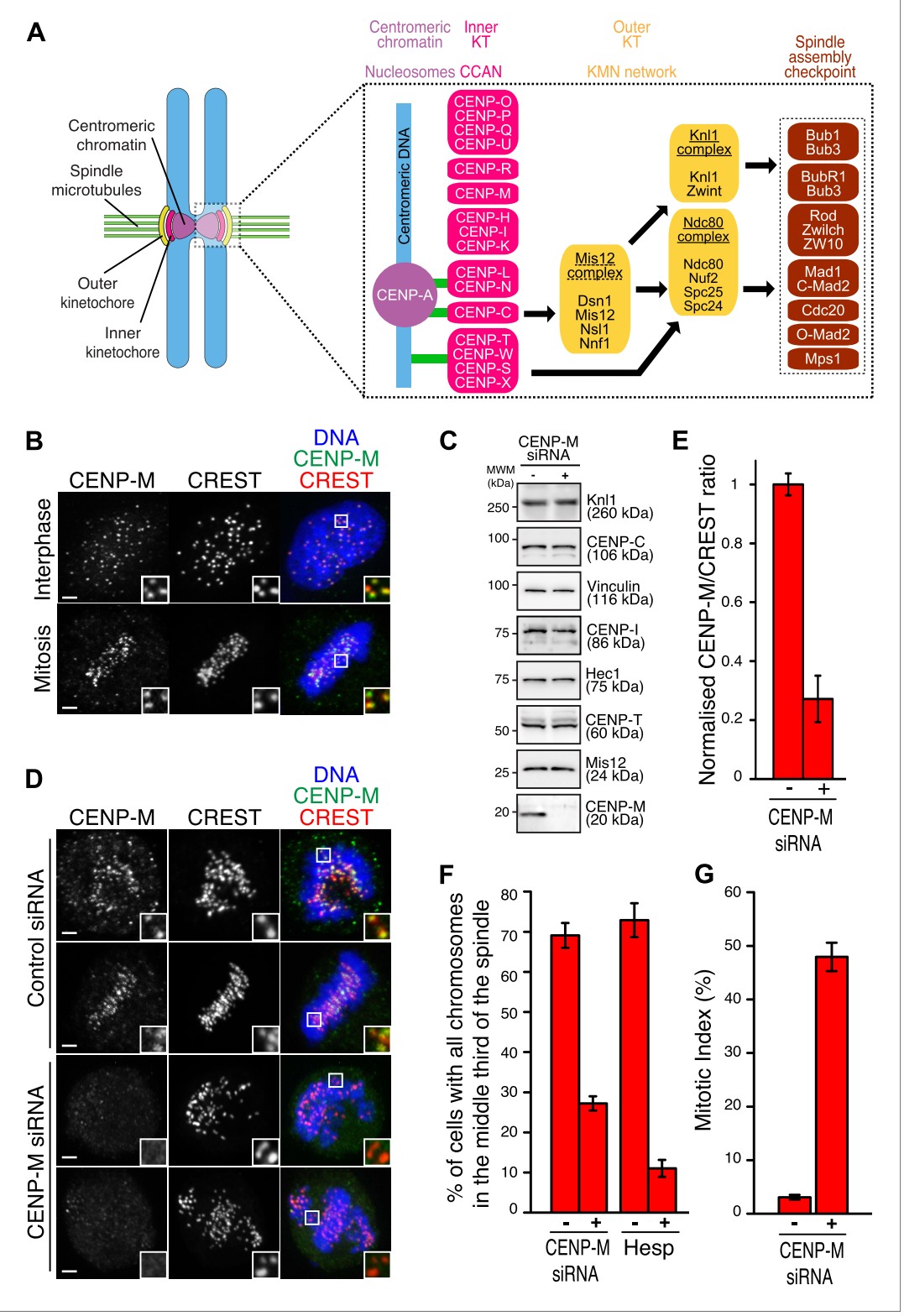

**Figure 1**. Ablation of CENP-M perturbs kinetochore function. (**A**) Core kinetochore components. CENP-C and CENP-T/W may create independent connections between centromeres and outer kinetochores. Green lines indicate direct connections with centromeric DNA or chromatin. Black lines indicate recruitment dependencies. CENP-C binds directly to CENP-A and Mis12 complex (*Przewloka et al., 2011*; *Screpanti et al., 2011*; *Kato et al., 2013*).
*Figure 1. Continued on next page*

*Figure 1. Continued*

CENP-T, together with CENP-W, S and X, may form a nucleosome-like structure interacting directly with DNA and the Ndc80 complex (*Hori et al., 2008a*; *Gascoigne et al., 2011*; *Schleiffer et al., 2012*; *Nishino et al., 2013*, *2012*). Sub-complexes of CCAN subunits were inferred from reconstitution or from similarity of depletion phenotypes (see main text). (**B**) Representative immunofluorescence (IF) images showing endogenous CENP-M localization to kinetochores of HeLa cells in both interphase and mitosis. Kinetochores were visualized with CREST sera and DNA stained with DAPI. Insets show a higher magnification of regions outlined by the white boxes. Scale bar = 2 μm. (**C**) Whole cell protein extracts from HeLa cells treated with specific siRNAs (showed in **D**) were run on SDS-PAGE and immunoblotted for the indicated kinetochore proteins. Vinculin was the loading control. MWM, molecular weight marker. (**D**) HeLa cells depleted for CENP-M display significant chromosome congression defects. Following fixation, cells treated with CENP-M siRNA were imaged for endogenous CENP-M, CREST and DNA (DAPI). Scale bars = 2 μm. (**E**) CENP-M kinetochore levels from the experiment in **D**. Quantifications are expressed as normalized CENP-M/CREST fluorescence intensity ratios. Graphs and bars indicate mean ± SEM. (**F**) Quantification of chromosome congression defects in **D**. As a positive control, cells treated with 500 nM Hesperadin were scored for alignment defects. (**G**) Quantification of the percentage of mitotic cells in the experiment in **D**.

The following figure supplements are available for figure 1:

**Figure supplement 1**. Additional localization data.

contains proteins with histone-fold domains that bind DNA and have been proposed to form a nucleosome-like structure (*Hori et al., 2008a*; *Nishino et al., 2012*). Together with CENP-C (*Earnshaw and Rothfield, 1985*), the CENP-T/W sub-complex has been shown to contribute to outer kinetochore assembly. CENP-C and CENP-T/W interact directly with the Mis12 and Ndc80 complexes, respectively (*Gascoigne et al., 2011*; *Przewloka et al., 2011*; *Screpanti et al., 2011*). CENP-C provides additional important scaffolding functions, as it binds directly to the CENP-A nucleosome and to the CENP-N/L complex (*Hinshaw and Harrison, 2013*; *Kato et al., 2013*). Several of these interactions among CCAN subunits have been recognized in *S. cerevisiae* and in *S. pombe*, suggesting a conserved plan of kinetochore assembly (*Cheeseman et al., 2002*; *Measday et al., 2002*; *De Wulf, 2003*; *Pidoux et al., 2003*; *Pot et al., 2003*; *Westermann et al., 2003*; *Liu et al., 2005*; *Tanaka et al., 2009*; *Westermann and Schleiffer, 2013*).

In this study, we concentrate on four additional CCAN subunits, CENP-H, CENP-I, CENP-K, and CENP-M. These proteins have been shown to be proximal to the subunits of the CENP-T/W/X/S, CENP-O/P/Q/U, and CENP-N/L complexes in cell lysates (*Obuse et al., 2004*; *Foltz et al., 2006*; *Izuta et al., 2006*; *Okada et al., 2006*), but their organization and pattern of interactions remain unclear (*Westhorpe and Straight, 2013*). Because similar chromosome alignment and kinetochore assembly defects were observed after their RNAi-based depletions, CENP-H, CENP-I and CENP-K were proposed to form a complex (*Okada et al., 2006*; *McClelland et al., 2007*; *Cheeseman et al., 2008*). Conversely, CENP-M was classified in a distinct phenotypic class, and tentatively assigned as a subunit of the CENP-L/N complex (*Okada et al., 2006*; *Westhorpe and Straight, 2013*). In this study, we elucidate the molecular basis of CENP-M's essential function in kinetochore assembly.

## Results

### Ablation of CENP-M perturbs kinetochore function

Antibodies against CENP-M or CENP-I stained interphase and mitotic kinetochores (*Figure 1B*, *Figure 1—figure supplement 1*), confirming that CENP-M and CENP-I reside constitutively at kinetochores (*Foltz et al., 2006*; *Okada et al., 2006*). To gain insight into CENP-M function, we depleted it by RNA interference (RNAi) in HeLa cells (*Figure 1C–E*). No change in the levels of several other kinetochore proteins was observed (*Figure 1C*). In mitotic HeLa cells, depletion of CENP-M prevented chromosome alignment at the metaphase plate to a comparable degree to that caused by Hesperadin, a small-molecule competitive inhibitor of Aurora B kinase, whose activity is crucially required for chromosome alignment (*Hauf et al., 2003*; *Figure 1F*). CENP-M depletion also caused robust mitotic arrest, likely a consequence of spindle checkpoint activation caused by chromosome alignment defects (*Figure 1G*). Thus, CENP-M is required for chromosome alignment and successful mitotic progression.

## CENP-M stabilizes CENP-I

To gain molecular insight on CENP-M, we purified recombinant CENP-M (*Figure 2A*) and asked if it interacted with recombinant versions of known kinetochore and centromere components in size-exclusion chromatography (SEC) co-elution experiments (in which proteins or protein complexes are separated on the basis of size and shape and co-elute if interacting). No interaction of CENP-M with a collection of recombinant proteins or protein complexes covering most known inner and outer kinetochore proteins or protein complexes was observed (summarized in *Table 1*).

CENP-I (known as Mis6 and Ctf3 in *S. pombe* and *S. cerevisiae*, respectively [*Saitoh et al., 1997*; *Measday et al., 2002*; *Nishihashi et al., 2002*]) had been initially excluded from these analyses because its expression in bacteria or insect cells had not resulted in a soluble product in cell lysates. Previous observations suggested that CENP-I might interact with CENP-H and CENP-K (*Pot et al., 2003*; *Okada et al., 2006*), but formal proof through reconstitution with recombinant proteins had been missing. CENP-H (known as Fta3 and Mcm16 in *S. pombe* and *S. cerevisiae*, respectively [*Sanyal et al., 1998*; *Sugata et al., 1999*; *Liu et al., 2005*]) and CENP-K (known as Sim4 and Mcm22 in *S. pombe* and *S. cerevisiae*, respectively [*Poddar et al., 1999*; *Pidoux et al., 2003*; *Foltz et al., 2006*; *Okada et al., 2006*]) (*Figure 2B*) form a tight dimer when co-expressed in, and purified from, insect cells (*Figure 2C*, *Figure 2—figure supplement 1*, panel A). Co-expression of full-length CENP-I or CENP-I$^{57-C}$ (a CENP-I fragment lacking the first 56 residues of CENP-I, predicted to be disordered) with CENP-H and CENP-K led to a partial solubilization of CENP-I, but the resulting soluble material was unstable and could not be purified homogenously (*Figure 2—figure supplement 1*, panel D). We therefore asked if further co-expression of CENP-M favored solubilization of CENP-I$^{57-C}$. Indeed, co-expression of CENP-M with CENP-I$^{57-C}$, CENP-H and CENP-K resulted in a soluble and stable 4-subunit complex that could be purified to homogeneity (*Figure 2D*, *Figure 2—figure supplement 1*, panels C–E). Thus, CENP-M stabilizes a quaternary complex containing CENP-H, CENP-I$^{57-C}$, CENP-K, and CENP-M, to which we refer as 'HIKM complex'. The theoretical molecular masses of the HIKM complex and of recombinant HI$^{57-C}$KM complex used in our studies are 166 and 159 kDa, respectively. In SEC experiments, the HI$^{57-C}$KM complex eluted close to the 158-kDa protein marker (*Figure 2D*), suggesting that the recombinant complex contains a single copy of each subunit.

## Subunit composition and interactions of the HIKM complex

To shed light on the organization of the HIKM complex, we subjected it to chemical cross-linking with the bifunctional reagent BS2G (bis[sulfosuccinimidyl]glutarate), which cross-links the primary amines of lysine side chains within a distance compatible with the length of the cross-linker (7.7 Å) (*Maiolica et al., 2007*; *Herzog et al., 2012*). Subsequent mass spectrometry analysis identified numerous cross-links between CENP-H and CENP-K, the majority of which mapped to the central domains of these proteins, which are likely to adopt an α-helical arrangement. The specific co-linear distribution of cross-links in this region suggests that CENP-H and CENP-K interact through an extended interface in a parallel arrangement (*Figure 2E*; the list of crosslinks is available in *Figure 2—source data 1*).

In addition to interacting with each other, the central domains of CENP-H and CENP-K displayed an extensive network of cross-links with the N-terminal region of CENP-I$^{57-C}$. Accordingly, we were able to reconstitute an interaction between a recombinant segment encompassing the N-terminal region of CENP-I (residues 57–281, which was expressed in a soluble form in the absence of other stabilizing components) and the CENP-H/K complex (*Figure 2F*), thus confirming that CENP-H/K binds the N-terminal region of CENP-I.

We did not observe cross-links between CENP-M and the CENP-H/K complex (*Figure 2E*), and indeed there was no detectable interaction between these proteins (*Figure 2—figure supplement 1*, panel F). Two cross-links, however, were observed between CENP-M and CENP-I (*Figure 2E*). In SEC experiments, CENP-M did not interact with CENP-I$^{57-281}$ (*Figure 2—figure supplement 1*, panel G) or with the CENP-HI$^{57-281}$K complex (*Figure 2G*), collectively suggesting that the incorporation of CENP-M in the CENP-HIKM complex requires CENP-I, and that the interaction of CENP-M with CENP-I requires residues located C-terminally to residue 281. However, co-expression of CENP-M with CENP-I was insufficient to solubilize CENP-I (*Figure 2—figure supplement 1*, left part of panel E). In summary, these observations suggest that CENP-I bridges CENP-H/K and CENP-M, and that both CENP-H/K and CENP-M contribute to CENP-I solubility and stabilization.

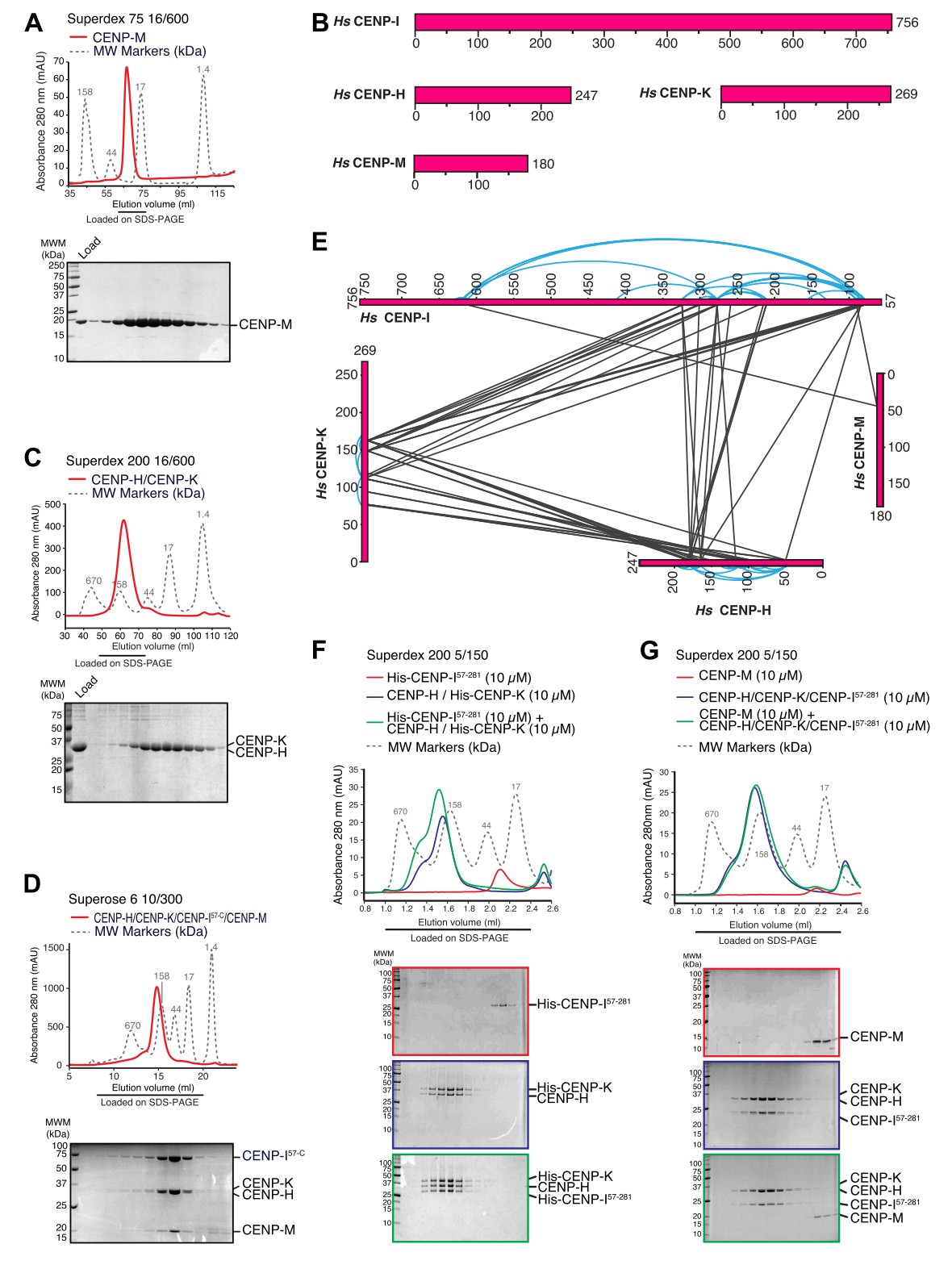

**Figure 2**. Characterization of the HIKM complex. (**A**) SEC elution profile of CENP-M with associated SDS-PAGE separations of peak fractions indicated by the horizontal bar under the profile. CENP-M (~20 kDa) elutes as expected for a monomeric species. (**B**) Schematic representation of the primary sequence of CENP-H, CENP-I, CENP-K, and CENP-M. (**C**) SEC elution profile and SDS-PAGE separation of the CENP-H/K complex. CENP-H/K forms a 1:1 dimer (~61 kDa) (*Figure 2—figure supplement 1*, panel **B**) but elutes near the 158 kDa marker, indicative of an elongated complex. (**D**) SEC elution

*Figure 2. Continued on next page*

*Figure 2. Continued*

profile and SDS-PAGE separation of the CENP-HI$^{57-C}$KM complex. CENP-HI$^{57-C}$KM (~159 kDa) elutes near the 158 kDa marker, suggesting that the complex contains a single copy of each subunit. (**E**) Summary of cross-links. Intra-molecular cross-links are shown in blue and outside the ideal perimeter designed by the four subunits of the complex. Inter-molecular cross-links are shown as black lines. (**F**) CENP-H/His-CENP-K complex and His-CENP-I$^{57-281}$, both at 10 µM, form a stoichiometric complex as shown by co-elution from SEC runs and corresponding SDS-PAGE separations. (**G**) Lack of co-elution from SEC runs and SDS-PAGE analysis indicate that CENP-H/K/I$^{57-281}$ complex and CENP-M do not bind.

The following source data and figure supplements are available for figure 2:

**Source data 1**.

**Figure supplement 1**. Additional biochemical characterization.

**Table 1.** List of potential binding interactions of CENP-M tested with purified proteins and complexes

| CENP-M incubation with: | Binding: |
| --- | --- |
| H3-containing mononucleosomes (DNA 601–167 bp) | NO |
| CENP-A-containing mononucleosomes (DNA 601–167 bp) | NO |
| CENP-C constructs (1–544, 509–760, 631–C-terminus) | NO |
| CENP-L/CENP–N complex | NO |
| CENP-H/CENP–K complex, CENP-I$^{57-281}$, CENP-H/CENP-K/CENP-I$^{57-281}$ complex | NO |
| CENP-O/CENP-P/CENP-Q/CENP–U complex, CENP-O/CENP-P and CENP-Q/CENP-U sub-complexes | NO |
| CENP-R | NO |
| CENP-T/CENP-W/CENP-S/CENP-X complex, CENP-T/CENP-W sub-complex (phosphorylated by Cdk1 or not), CENP-S/CENP-X sub-complex | NO |
| Mis12 complex | NO |
| Ndc80 complex | NO |
| Knl1$^{2000-2311}$ | NO |
| Zwint | NO |
| KMN network | NO |
| Microtubules | NO |

## Structural analysis reveals a small G-protein fold in CENP-M

We generated crystals of residues 1–171 of CENP-M (a fragment identified by limited proteolysis) (**Figure 3—figure supplement 1**, panel A) and determined their crystal structure by the SAD (single-wavelength anomalous diffraction) method to a resolution of 1.5 Å using a SeMet derivative (**Table 2**). The structure of CENP-M$^{1-171}$ is globular and consists of a five-stranded parallel β-sheet surrounded by six α-helices (**Figure 3A**, **Figure 3—figure supplement 1**, panel B). A search of the Protein Data Bank (PDB) using the Dali server (**Holm and Rosenström, 2010**) indicated that CENP-M is structurally related to small GTPases (for comparison, the structure and topology of the small GTPase Rab1 are shown in **Figure 3B** and **Figure 3—figure supplement 1**, panel C, respectively), as previously predicted in a bioinformatics analysis (**Santaguida and Musacchio, 2009**; **Westermann and Schleiffer, 2013**).

Small GTPases are guanine nucleotide binding proteins that control a variety of essential cellular functions (**Vetter and Wittinghofer, 2001**; **Cherfils and Zeghouf, 2013**). They can act as molecular switches by engaging in different interactions with effectors that depend on the bound nucleotide (GTP or GDP). Small GTPases describe a superfamily of proteins, which are generally classified in several families (**Rojas et al., 2012**). Structural superposition with representative members of different families identified Rab proteins as the closest structural homologs of CENP-M (**Figure 3—figure supplement 2**). A sequence alignment derived from the structural superposition of HsCENP-M with HsRab1A (PDB code 4FMC, [**Dong et al., 2012**]) shows that CENP-M has lost essential sequence motifs required for GTP binding and hydrolysis by small G-proteins (**Vetter and Wittinghofer, 2001**; **Figure 3C**). For instance, the glycine-rich P-loop (GX$_4$GKS/T, where X is any aminoacid), which is involved in GTP binding by GTPases, lacks two essential Gly residues in CENP-M. The so-called switch I and switch II regions of small GTPases, which are sensitive to nucleotide hydrolysis and create a dependency on nucleotide status for the interaction of GTPases with their effectors (**Vetter and Wittinghofer, 2001**; **Cherfils and Zeghouf, 2013**), are also profoundly modified in CENP-M. A deletion of the entire α1-β2 loop and the β2 strand effectively removed the switch I region (**Figure 3C**), while the sequence corresponding to switch II (DxxG motif) is highly divergent. Thus, HsCENP-M has the fold of a small GTPase, but may be unable to bind (and

**Table 2.** Data collection, phasing and refinement statistics

| | Native | Derivative |
|---|---|---|
| Data collection | | |
| Beamline | ESRF ID14-4 | SLS X06DA (PXIII) |
| Spacegroup | P3 | P3 |
| Unit cell parameters (Å, °) | a = b = 104.50, c = 33.59 | a = b = 104.03, c = 33.56 |
| | α = β = 90, γ = 120 | α = β = 90, γ = 120 |
| Wavelength (Å) | 0.91970 | 0.97942 |
| Resolution limits (Å) | 52.25–1.49 (1.54–1.49)* | 31.45–2.00 (2.06–2.00)* |
| Reflections observed/unique | 607786/64606 | 419456/27271 |
| Completeness (%) | 98.3 (96.3)* | 99.9 (98.9)* |
| $R_{sym}$†(%) | 5.6 (36.1)* | 9.4 (75.2)* |
| $<I>/<\sigma I>$ | 26.3 (6.4)* | 24.1 (3.9)* |
| Redundancy | 9.4 (8.8)* | 15.4 (13.9)* |
| SAD phasing | | |
| BAYES-CC | | 49.1 ± 18.5 |
| Se sites found/expected | | 5/6 |
| FOM before solvent flattening and density modification | | 0.35 |
| FOM after solvent flattening and density modification | | 0.69 |
| Refinement | | |
| Resolution limits (Å) | 52.25–1.49 (1.52–1.49)* | |
| Reflections for $R_{cryst}$/for $R_{free}$ | 59806/4800 | |
| $R_{cryst}$‡(%) | 12.4 (20.6)* | |
| $R_{free}$‡(%) | 16.4 (23.0)* | |
| No. of protein atoms/water atoms | 2307/297 | |
| Average B factor protein atoms/water atoms (Å$^2$) | 21.67/35.34 | |
| RMSD bond lengths (Å) | 0.005 | |
| RMSD bond angles (°) | 0.854 | |
| Twin fraction (operator −h, −k, l) | 0.49 | |
| Ramachandran Plot Statistics§ | | |
| Favoured region (%) | 99.3 | |
| Outliers (%) | 0.0 | |

BAYES-CC: Bayesian estimate of the correlation coefficient (CC) between the experimental map and an ideal map, reported as CC * 100 ± 2 standard deviations.
FOM: figure of merit.
RMSD: root mean square deviation.
*Values in parentheses refer to the highest resolution shell.
†$R_{sym} = \Sigma_h\Sigma_i|I_{h,i} − <I_h>|/\Sigma_h\Sigma_i\ I_{h,i}$.
‡$R_{cryst}$ and $R_{free} = \Sigma|F_{obs} − F_{calc}|/\Sigma\ F_{obs}$; $R_{free}$ calculated for a 7.4% subset of reflections not used in the refinement.
§Calculated using MOLPROBITY within the PHENIX suite.

therefore hydrolyze) GTP. Experiments with MANT nucleotides (*Hiratsuka, 1983*) confirmed this prediction directly (*Figure 3D*, *Figure 3—figure supplement 1*, panel D).

Despite the divergence of CENP-M from *bona fide* GTPases, several elements indicate that CENP-M evolved from active GTPases. For instance, distant CENP-M family members retain a canonical P-loop (*Figure 3—figure supplement 1*, panel E, 'Discussion'). More importantly, phylogenetic analyses using a CENP-M/Rab1 structure-based sequence alignment (*Figure 3E*, see 'Materials and methods' for details) replicated the distribution of Ras sub-families previously described using a different template alignment (*Rojas et al., 2012*). CENP-M family members cluster at the base of the RAB/RAS

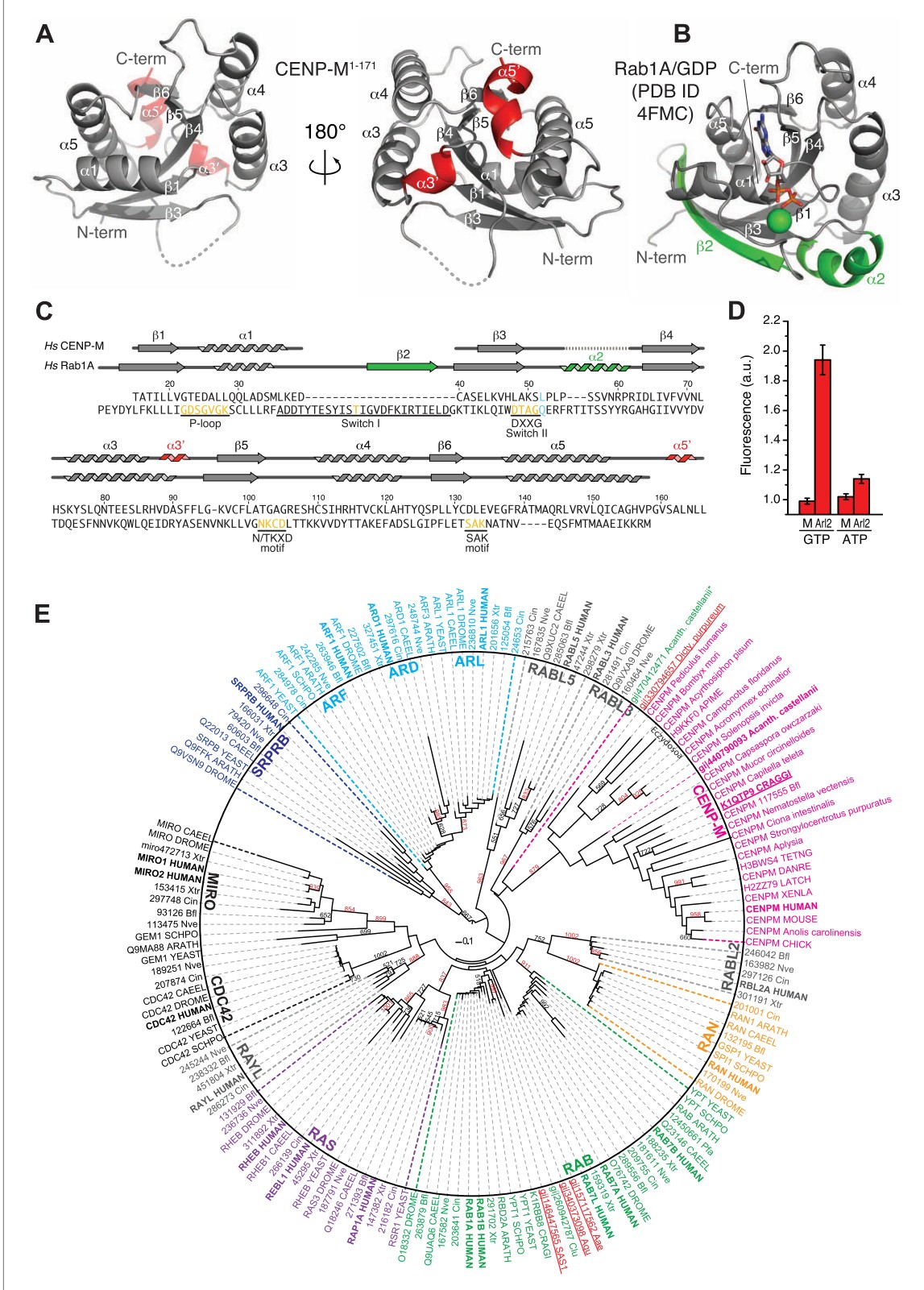

**Figure 3**. A small G-protein fold in CENP-M. (**A**) Cartoon model of CENP-M[1–171] in two orientations. (**B**) Cartoon model of Rab1A/GDP (PDB ID 4FMC). (**C**) Sequence alignment based on the structural superposition of CENP-M[1–171] with Rab1A. Conserved elements of small G proteins are in yellow. A conserved residue involved in catalysis and targeted by activating mutations in Ras is in light blue. Secondary structure elements of CENP-M[1–171] not

*Figure 3. Continued on next page*

*Figure 3. Continued*

present in Rab1A/GDP are in red, while those present in Rab1A/GDP but not in CENP-M[1–171] are in green. (**D**) Experiments with N-methylanthraniloyl (MANT) derivatives of GTP and ATP. Binding of MANT-GTP or MANT-ATP to the small GTPase Arl2 ('Arl2') or CENP-M ('M') was monitored at an emission wavelength of 440 nm (See *Figure 3—figure supplement 1*, panel **D**). The histogram shows the time-averaged fluorescence value after addition of the indicated proteins to a solution of the indicated MANT nucleotides, normalized against the time-averaged value prior to protein addition. Only the addition of Arl2 to MANT-GTP (or MANT-GDP, see *Figure 3—figure supplement 1*, panel **D**) gave a clear increase in signal indicative of a physical interaction of the MANT nucleotide with Arl2. (**E**) Unrooted maximum likelihood tree of 157 sequences. Shown are members of classical small GTPase families in many species, covering a wide range of evolutionary points. Gray names are reclassified families (*Rojas et al., 2012*). Bold names are human sequences. CENP-M sequences are pink. Uppercase indicates Uniprot code for proteins. 3-code labels are: Nve (*Nematostella vectensis*), Bfl (*Branchiostoma floridae*), Xtr (*Xenopus tropicalis*), and Cin (*Ciona intestinalis*). Numbers on the left of 3-code labels are accession numbers corresponding to the DOE Joint Genome Institute (JGI) database. Red and underlined names are Genbank gi identifiers found in iterative hmmer searches against non-redundant database. Acanth. castellanii indicates *Acantanthamoeba castellanii*, where * indicates that this entry is annotated as a RAS protein. Dicty. purpureum is *Dictyostelium purpureum*, Aae is *Aedes aegypti*, Aqu is *Amphimedon queenslandica* (sponge), and Clu is *Clavispora lusitaniae* (fungi). Red numbers within the tree indicate number of trees corresponding to more than 80% of statistical support for a given group, whereas black indicates values below 80%. Only representative numbers have been shown for clarity.

The following figure supplements are available for figure 3:

**Figure supplement 1**. Additional analyses of CENP-M (related to *Figure 3*).

**Figure supplement 2**. Comparison of CENP-M with Rab-like GTPases.

---

sub-families, suggesting that CENP-M might have originated at a similar time. The relative location of the ARF/SRPRB sub-families seems to exclude CENP-M as a basal member of the superfamily. Early divergent eukaryotes (such as amoebas) and early metazoans (molluscs) express both CENP-M and *bona fide* GTPases. CENP-M, however, has been lost from most fungi, where additional protein products of specific lineage expansions or existing Rab proteins may fulfil its role ('Discussion'). In conclusion, we identify CENP-M as a 'pseudo G-protein', in analogy to inactive kinase domains, indicated with the term 'pseudokinase'.

## Structural organization of the HIKM complex

Because the structure of CENP-I is unknown, we submitted its sequence to the structure prediction servers I-TASSER, Phyre2 and Rosetta (*Das and Baker, 2008*; *Zhang, 2008*; *Kelley and Sternberg, 2009*). The α-solenoid fold of β-karyopherins such as Importin-β was consistently identified as a high-confidence template for CENP-I structural modeling (*Figure 4A*). Similar results were obtained when structural predictions were carried out with the sequences of CENP-I homologs, including Mis6 (*S. pombe*) and Ctf3 (*S. cerevisiae*) (*Figure 4—figure supplement 1A–B*). The structure of Importin-β consists of a tandem series of HEAT (Huntingtin, elongation factor 3, PR65/A subunit of protein phosphatase 2A and kinase TOR) repeats, helical hairpins that stack against each other to create a twisted super-helical arrangement. We could not unequivocally demonstrate the existence of HEAT repeats in CENP-I. However, program RADAR (*Heger and Holm, 2000*) predicted the presence of repeats within the presumed α-helical part of the protein (*Figure 4—figure supplement 1C*).

Lysines clustering near the N- and C-termini of CENP-I are involved in numerous intra-CENP-I cross-links (*Figure 2E*), suggesting that the N- and C-terminal regions of CENP-I are in relatively close proximity, possibly reflecting a super-helical arrangement. Furthermore, the only two cross-links between CENP-M and CENP-I mapped near the N- and the C-termini of CENP-I (*Figure 2E*). Thus, we speculate that CENP-M may bind, becoming largely buried, near the concave surface of the predicted CENP-I solenoid, in a manner that is reminiscent of the interaction of β-importin with the small GTPase Ran (*Vetter et al., 1999*; *Figure 4A*). In agreement with this hypothesis, there is strong sequence conservation near the surface of the I-TASSER model of CENP-I predicted to interact with CENP-M (*Figure 4—figure supplement 2A,B*).

We determined the three-dimensional (3D) structure of the HIKM complex by negative stain electron microscopy and single particle analysis (*Figure 4B,C*). The resulting map (*Figure 4C*) extended to a maximal resolution of 22 Å (*Figure 4—figure supplement 3*). The HIKM complex has a long axis of ~14 nm and short axes of ~5–7 nm, and a rather irregular shape, with a broader 'base' and a 'head' domain from which a prominent 'nose' domain emerges. The model of the CENP-I/M complex fits

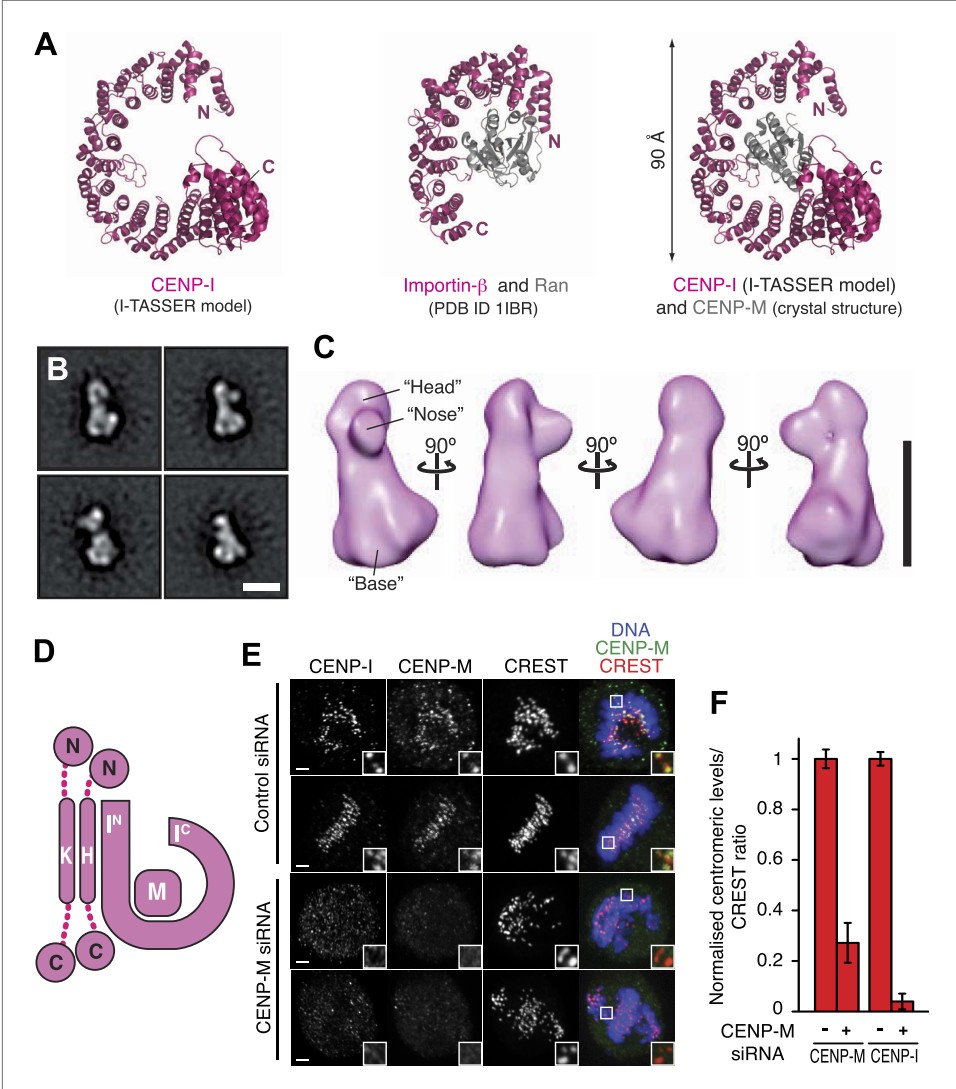

**Figure 4**. Structural organization of the HIKM complex. (**A**) Cartoon representation of the CENP-I model generated by program I-TASSER (left), of the Importin-β/Ran complex (middle), and of a hypothetical structure between CENP-I and CENP-M modeled on the Importin-β/Ran complex (right). A scoring function (C-score) associated with I-TASSER models estimates accuracy of structure predictions. C-score is typically in a range from −5 to 2, where a higher score reflects a model of better quality. Both false positive and false negative rates are estimated to be below 0.1 when a C-score >−1.5 is displayed (**Zhang, 2008**). The CENP-I model is associated with a C-score of −1. (**B**) Representative class averages of the negatively stained HIKM complex. **Figure 4—figure supplement 3** shows the complete set of class averages. Scale bar = 10 nm. (**C**) A 3D reconstruction of HIKM complex from negatively stained particles at ~22 Å resolution. Scale bar = 10 nm. (**D**) Summary of interactions in the CENP-HIKM complex. The central regions of CENP-H and CENP-K may form an extended parallel interaction, possibly through an α-helical arrangement, which interacts more or less co-linearly with the N-terminal region of CENP-I (I$^N$). Additional globular domains may be present at the N- and C-termini of CENP-H and CENP-K. The entire sequence of CENP-I may fold as a helical solenoid. CENP-M does not interact with CENP-H/K and may bind near the concave surface of the predicted CENP-I solenoid, becoming largely buried. (**E**) siRNA depletion of endogenous CENP-M abrogates CENP-I kinetochore localization in HeLa cells. Representative cells displayed here are the same shown in **Figure 1D**, but with addition of CENP-I staining (left panels). Insets display a higher magnification of regions outlined by white boxes. Scale bars = 2 μm. (**F**) CENP-M and CENP-I kinetochore levels from the experiment illustrated in **E**. Quantification for CENP-M kinetochore levels are the same shown in **Figure 1D** and were performed as previously described. Graphs and bars indicate mean ± SEM.

*Figure 4. Continued on next page*

*Figure 4. Continued*

The following figure supplements are available for figure 4:

**Figure supplement 1**. Structural predictions on CENP-I orthologs.

**Figure supplement 2**. Conservation mapped on the CENP-I model.

**Figure supplement 3**. EM analysis.

**Figure supplement 4**. Fitting the CENP-I/M model in the EM density.

snugly in the base domain of this density (*Figure 4—figure supplement 4*), suggesting that CENP-H/K occupy the head and nose domains. Our future studies will address the validity of this fitting. *Figure 4D* summarizes the hypothetical topological organization of the HIKM complex emerging from the battery of structural and biochemical analyses reported above.

## CENP-M is required for kinetochore targeting of CENP-I

Collectively, our observations identify a crucial role of CENP-M in the stabilization of CENP-I. We therefore asked if CENP-M is required for kinetochore localization of CENP-I. In line with this expectation, RNAi-based depletion of CENP-M resulted in the complete disappearance of CENP-I from kinetochores (*Figure 4E,F*).

In principle, the loss of CENP-I from kinetochores upon CENP-M depletion might reflect the loss of the interaction between CENP-M and CENP-I, but also the loss of additional interactions required for kinetochore stability. To overcome this objection, we sought to identify point mutations in CENP-M affecting its interaction with CENP-I. To identify such mutants, we focused on conserved residues exposed at the surface of CENP-M (indicated with an asterisk in *Figure 5A* and displayed in *Figure 5B*; see also the alignment in *Figure 3—figure supplement 1*, panel E), reasoning that at least a subset of such conserved surface-exposed residues might be involved in CENP-I binding. We then co-expressed GST-CENP-M or its point mutants with CENP-H, CENP-I, and CENP-K in insect cells, and monitored the amount of CENP-I co-purifying with GST-CENP-M on a glutathione–sepharose resin (data not shown). A double point mutant of GST-CENP-M, GST-CENP-M$^{L94A-L163E}$, was unable to precipitate CENP-H, CENP-I, and CENP-K when co-expressed in insect cells, contrarily to GST-CENP-M$^{wt}$ (*Figure 5C*). Mutation of L94 and L163 did not affect the solubility of CENP-M, nor its behavior during the purification procedure (*Figure 5—figure supplement 1*).

To assess the effect of these mutations in cells, GFP-CENP-M$^{wt}$ or GFP-CENP-M$^{L94A-L163E}$ were expressed from an inducible promoter after stable integration in HeLa cells. Precipitates of GFP-CENP-M$^{wt}$ contained CENP-I, the CCAN subunits CENP-T/W, and the KMN protein Mis12. Conversely, GFP-CENP-M$^{L94A-L163E}$ was unable to establish any of these interactions (*Figure 5D*). Thus, the CENP-I binding surface of CENP-M is required to establish interactions with other kinetochore proteins. RNAi-resistant GFP-CENP-M$^{wt}$ localized to kinetochores (*Figure 5—figure supplement 2*, panels A,B) and rescued the severe chromosome alignment defect caused by the depletion of endogenous CENP-M by RNAi, as well as kinetochore localization of CENP-I, demonstrating the specificity of the RNAi-based depletion of CENP-M. GFP-CENP-M$^{L94A-L163E}$, on the other hand, did not rescue chromosome alignment or CENP-I localization (*Figure 5E,F*). Furthermore, GFP-CENP-M$^{L94A-L163E}$ did not localize to kinetochores (*Figure 5—figure supplement 2*, panel D), indicating that CENP-M and CENP-I are mutually required for kinetochore localization. Collectively, our analysis highlights the importance of CENP-M as a stabilization factor for CENP-I in vitro and in vivo.

## Significance of the CENP-M/CENP-I interaction for kinetochore assembly

Next, we wished to study the CENP-M/I interaction in the context of inner and outer kinetochore assembly (*Figure 1A*), focusing in particular on CENP-T and CENP-C. Both these proteins likely span a large fraction of the physical distance that separates chromatin from microtubules (*Hori et al., 2008a, 2013*; *Wan et al., 2009*; *Gascoigne et al., 2011*; *Przewloka et al., 2011*; *Screpanti et al., 2011*; *Suzuki et al., 2011*; *Schleiffer et al., 2012*; *Nishino et al., 2013*). CENP-C interacts directly

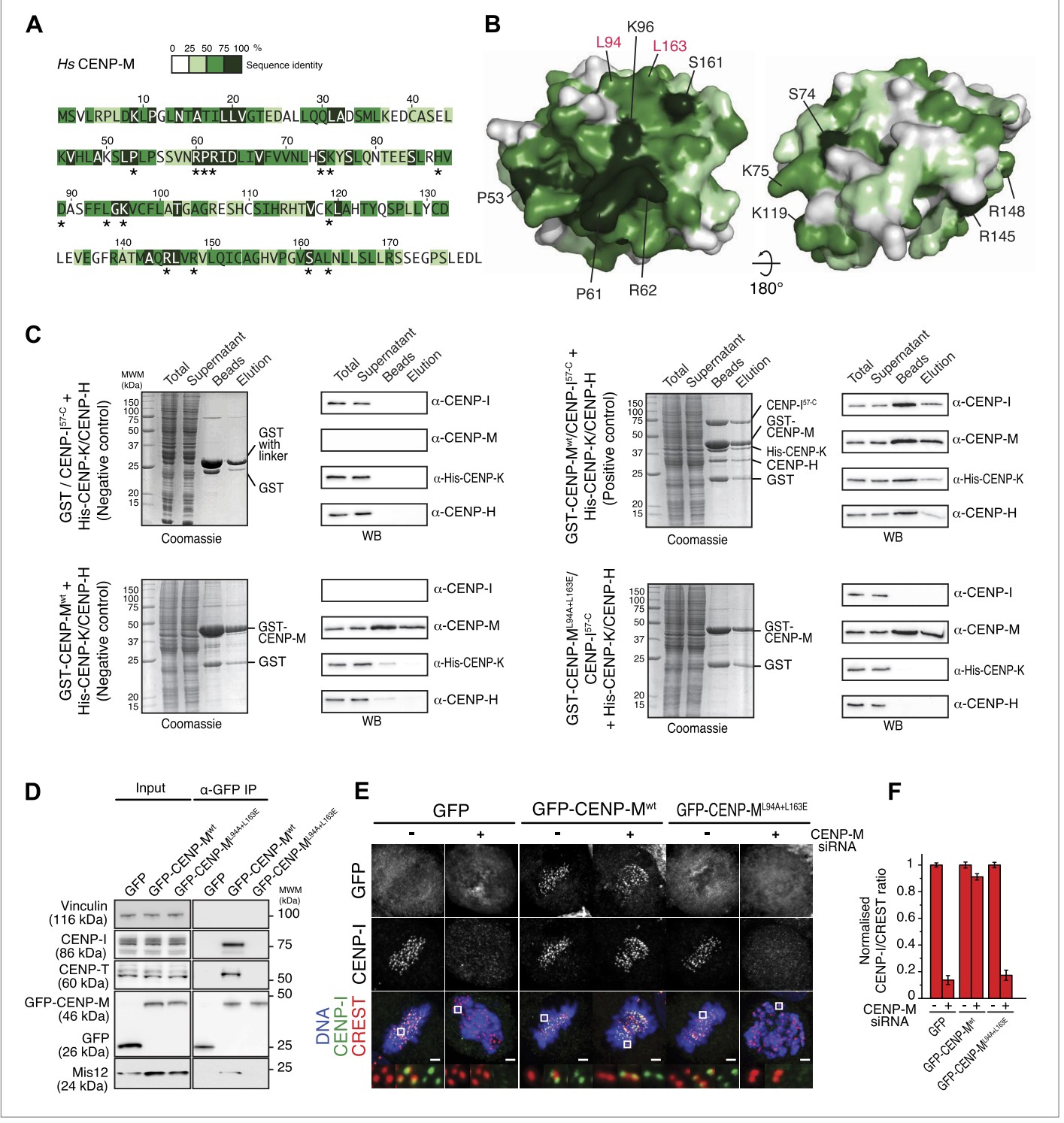

**Figure 5**. CENP-M residues required for kinetochore targeting of CENP-I. (**A**) The CENP-M alignment identifies highly conserved residues (based on alignment in *Figure 3—figure supplement 1*, panel **D**), a subset of which (asterisks) is exposed at the surface of CENP-M. (**B**) Position of conserved residues shown in **A** on two opposite faces of the CENP-M surface. (**C**) After insect cell co-expression of indicated proteins, affinity purification with GST-CENP-M (if present) led to isolation of associated proteins shown, after SDS-PAGE separation, in the 'beads' fraction. Material eluted from beads was collected and shown in lanes labeled 'elution'. Co-expression of all four CENP-HI57–CKM subunits (positive control) is necessary for identification of the CENP-HI57–CKM complex on beads and in elution fraction. CENP-ML94A–L163E fails to assemble CENP-HI57–CKM despite the expression of CENP-H, CENP-I57–C and CENP-K. (**D**) GFP-CENP-Mwt, but not GFP-CENP-ML94A+L163E, co-immunoprecipitates CENP-I, CENP-T and Mis12 from mitotic cells. Panels

*Figure 5. Continued on next page*

*Figure 5. Continued*

represent the α-GFP co-immunoprecipitation analysis of protein extracts obtained from mitotic HeLa Flp-In T-REx cells stably expressing GFP, GFP-CENP-M^wt or GFP-CENP-M^L94A+L163E from an inducible promoter. Total protein extracts (Input) and immunoprecipitates (α-GFP IP) were run on SDS-PAGE and subjected to WB with indicated antibodies. Vinculin was used as a loading control. (**E**) Representative images of HeLa Flp-In T-REx cells treated with siRNA for endogenous CENP-M and expressing the indicated siRNA-resistant GFP-CENP-M fusions. Expression of GFP-CENP-M^wt, but not of GFP-CENP-M^L94A+L163E, rescues chromosome alignment defects and loss of CENP-I kinetochore localization observed upon depletion of endogenous CENP-M. Following fixation, cells were immunostained and imaged for GFP, CENP-I, CREST and DNA (DAPI). Insets show a higher magnification of regions outlined by white boxes. Scale bars = 2 μm. (**F**) Quantification, for experiment in **E**, of the CENP-I kinetochore levels normalized to CREST kinetochore signal. Graphs and bars indicate mean ± SEM. See 'Materials and methods' section for details on quantification.

The following figure supplements are available for figure 5:

**Figure supplement 1**. Stability of CENP-M mutant.

**Figure supplement 2**. Inducible expression and localization of CENP-M.

---

with the CENP-A nucleosome (*Carroll et al., 2010*; *Kato et al., 2013*). CENP-T, on the other hand, has been proposed to form a nucleosome-like structure that might flank the CENP-A nucleosome and other canonical H3 nucleosomes in centromeres (*Hori et al., 2008a*; *Nishino et al., 2012*; *Takeuchi et al., 2014*). Mutual independence in their kinetochore recruitment has led to suggest that CENP-C and CENP-T contribute two major independent axes for outer kinetochore assembly (*Hori et al., 2008a, 2013*; *Gascoigne et al., 2011*). Indeed, both CENP-T and CENP-C interact directly with outer kinetochore components, and the CENP-T/W complex has been proposed to contribute to a pathway of Ndc80 recruitment that is independent of the Mis12 complex (*Gascoigne et al., 2011*; *Schleiffer et al., 2012*; *Hori et al., 2013*).

Depletion of CENP-M did not perturb the kinetochore levels of CENP-C (*Figure 6A*; see *Figure 6E* for quantification of fluorescence intensity data for panels A–D). CENP-C has been implicated in a direct interaction with the Mis12 complex (*Przewloka et al., 2011*; *Screpanti et al., 2011*; *Figure 1A*). Consistently with the retention of CENP-C in cells depleted of CENP-M, the kinetochore levels of Nsl1, a subunit of the Mis12 complex, remained largely normal after CENP-M depletion (*Figure 6B*). Depletion of CENP-M, on the other hand, resulted in co-depletion of CENP-T/W from kinetochores. Importantly, this defect was rescued by GFP-CENP-M^wt but not CENP-M^L94A–L163E (*Figure 6C*). In line with the idea that CENP-T/W contributes to recruit the Ndc80 complex to kinetochores, loss of CENP-T/W in CENP-M depleted cells correlated with a severe reduction of the kinetochore levels of Ndc80 (also known as Hec1), a subunit of the Ndc80 complex (*Figure 6D*). Collectively, these results suggest that reduced kinetochore levels of CENP-T/W in cells depleted of CENP-M reduce Ndc80 localization and generate chromosome alignment defects.

Previously, kinetochore localization of at least a subset of HIKM subunits has been shown to depend on CENP-C (*Milks et al., 2009*; *Carroll et al., 2010*; *Gascoigne et al., 2011*). Because kinetochore localization of CENP-T/W depends on the HIKM complex (*Figure 6C*), we therefore hypothesized that kinetochore accumulation of CENP-T/W might also rely on CENP-C. Indeed, depletion of CENP-C prevented kinetochore localization of CENP-T/W both in interphase (*Figure 6F*; quantified in *Figure 6G*) and in mitotis (not shown), whereas depletion of CENP-T/W did not alter CENP-C localization (*Figure 6F,G*). These results are in agreement with a previous report (*Carroll et al., 2010*) and highlight the essential role of CENP-C as the basis of the pathway of kinetochore assembly of the HIKM and CENP-T/W complexes.

## The HIKM complex interacts with CENP-T/W

Finally, we asked if the loss of CENP-T/W from kinetochores upon depletion of CENP-M could reflect a direct interaction between these proteins. We immobilized GST (as control) or a GST-tagged version of the HIKM complex (containing GST-CENP-M) on solid phase as baits, and exposed them to different prays. Untagged CENP-T/W/X/S complex (*Nishino et al., 2012*) bound directly to the HIKM complex and was retained on GST-HIKM beads (*Figure 7A*. Note that the band corresponding to CENP-X is rather diffuse and poorly visible). A version of the CENP-T/W/S/X complex containing CENP-T^458–C, which only contains the histone-fold domain of CENP-T, rather than full-length CENP-T, bound to GST-HIKM equally effectively (*Figure 7B*). Indeed, the CENP-T^458–C/W complex was sufficient for an

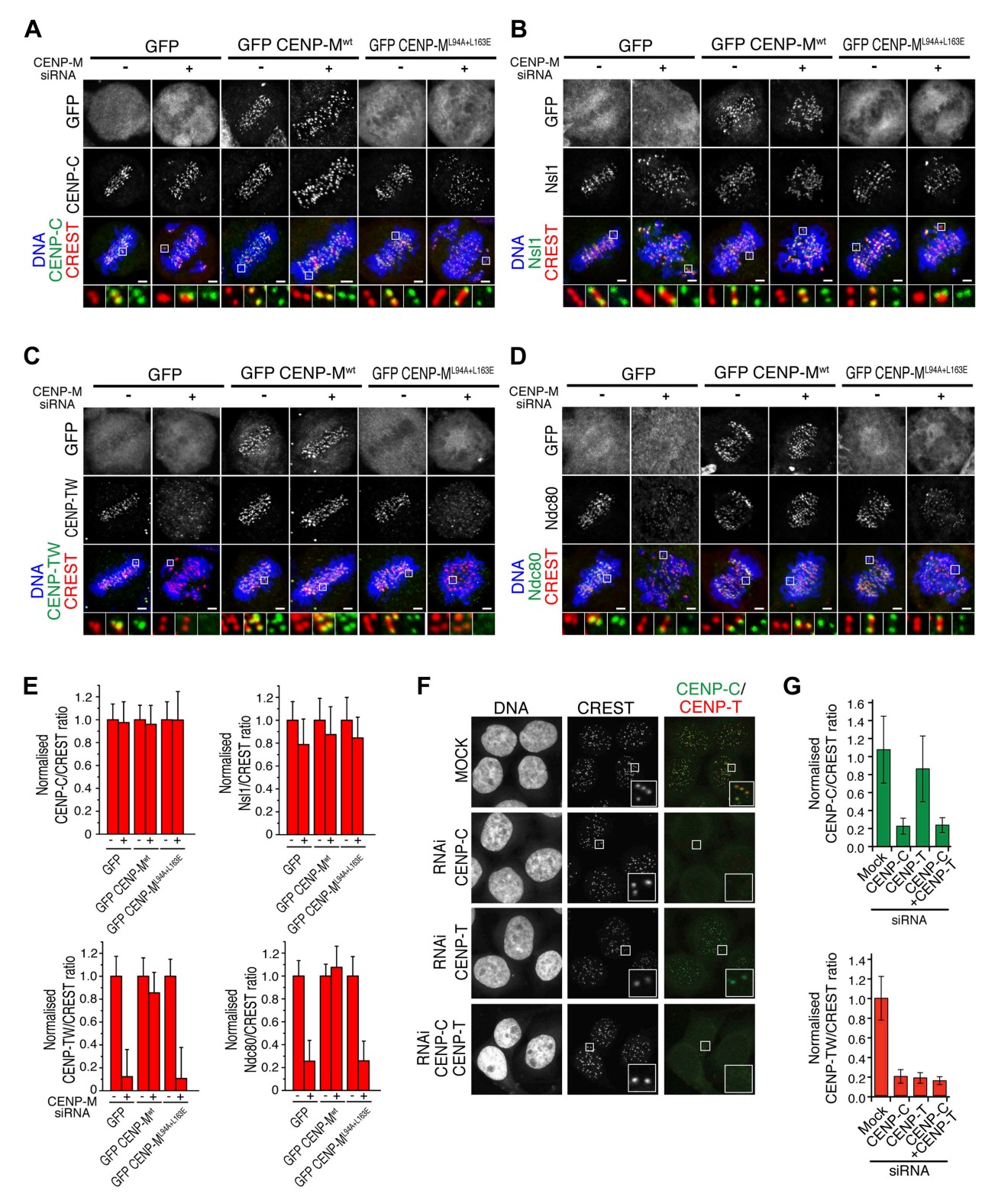

**Figure 6**. Significance of the CENP-M/CENP-I interaction for kinetochore assembly. (**A–D**) Representative images of the localization of kinetochore proteins in HeLa Flp-In T-REx cells treated with siRNA for endogenous CENP-M and expressing the indicated siRNA-resistant GFP-CENP-M fusions. Scale bars = 2 μm. (**E**) Quantification, for experiments **A–D**, of the kinetochore levels of the indicated proteins normalized to CREST. Graphs and bars indicate mean ± SEM. (**F**) Depletion of CENP-C abrogates kinetochore accumulation of CENP-T/W. Representative images of HeLa cells treated with

*Figure 6. Continued on next page*

*Figure 6. Continued*

siRNA for CENP-C or CENP-T and arrested in G2 with the Cdk1 inhibitor RO-3306 ('Materials and methods'). Following fixation, cells were immunostained for CENP-C, CENP-T/W and CREST. DNA was stained with DAPI. Scale bars = 10 µm. (**G**) Quantification, for experiment **F**, of the kinetochore levels of the indicated proteins normalized to CREST kinetochore signal. Graphs and bars indicate mean ± SEM.

interaction with GST-HIKM (*Figure 7C*), whereas no binding to GST-HIKM was observed with CENP-S/X in the absence of CENP-T/W (*Figure 7D*). Collectively, these results clarify that the CENP-T/W complex interacts directly with the HIKM complex, whose subunits are required for the recruitment of the CENP-T/W complex to kinetochores.

## Discussion

CENP-M was initially identified for its high expression in proliferating cells, and named accordingly PANE1, for proliferation associated nuclear element 1 (*Bierie et al., 2004*). Subsequently, CENP-M was shown to be closely associated with CENP-A, as well as with CENP-H, CENP-K, CENP-I, CENP-L, CENP-N, and CENP-T (*Obuse et al., 2004*; *Foltz et al., 2006*; *Izuta et al., 2006*; *Okada et al., 2006*). RNAi-based depletion of CENP-M caused mis-localization of other CCAN subunits, thus pointing to an important role of CENP-M in inner kinetochore stability (*Foltz et al., 2006*; *Okada et al., 2006*). However, the molecular mechanisms subtending to the function of CENP-M and the physical interactions in which CENP-M engages at the kinetochore had remained unknown.

Kinetochores, like many other cellular structures, are patchworks of different protein–protein interaction motifs and domains. For instance, at least six kinetochore proteins, including the CCAN subunit CENP-O and CENP-P (respectively orthologs of Mcm21 and Ctf19 of *S. cerevisiae*), contain RWD domains (*Schmitzberger and Harrison, 2012*; *Petrovic et al., 2014*). More recently, the structure of the Chl4/Iml3 complex of *S. cerevisiae*, respectively orthologs of CENP-N and CENP-L, revealed structural similarity with the bacterial recombination-association protein RdgC and with TATA-binding protein TBP (*Guo et al., 2013*; *Hinshaw and Harrison, 2013*). In this study, we have extended the collection of kinetochore folds by demonstrating that CENP-M folds like a small GTPase, as recently postulated based on prediction methods (*Westermann and Schleiffer, 2013*). CENP-M is devoid of all essential sequence signatures associated with nucleotide binding, hydrolysis and conformational switching of small GTPases (*Vetter and Wittinghofer, 2001*; *Rojas et al., 2012*; *Cherfils and Zeghouf, 2013*). An invariant feature of CENP-M is the presence of leucine at position 52 (of HsCENP-M), equivalent to Gln61 (Q61) at the end of the Switch II region of Ras (*Figure 3C*). Mutation of Gln61 to Leu impairs GTP hydrolysis (*Scheidig et al., 1999*) and unleashes the transforming potential of Ras. Thus, the first step leading CENP-M to diverge from genuine GTPases might have been an impairment of GTP hydrolysis, followed by the loss of other features associated with GTP handling. Consistent with this hypothesis, a substantially conserved P-loop is present in distant CENP-M family members (*Figure 3—figure supplement 1*, panel E). Our phylogenetic analysis identified CENP-M as a *bona fide* member of the small GTPase tree and suggests that CENP-M might have evolved from an ancestor shared with Rab-family GTPases.

Interestingly, CENP-M can only be identified in metazoans but not in fungal genomes. Conversely, the other components of the complex, CENP-H, CENP-I, and CENP-K, are nearly ubiquitously conserved and are clearly identified in fungi, where they appear to interact (*Measday et al., 2002*). Given the importance of CENP-M in kinetochore assembly and stability, and its interaction with evolutionarily conserved proteins, its absence in several representatives of Opisthokonta is puzzling. We consider it unlikely that the fission yeast protein Mis17 is a CENP-M ortholog, as recently proposed (*Shiroiwa et al., 2011*), because all computational prediction tools we tested failed to identify a domain related to GTPases in Mis17 (data not shown). It is possible, however, that Mis17 acts as a functional analogue of CENP-M. It is also legitimate to speculate that a functional GTPase, possibly a Rab-family GTPase given its evolutionary proximity to CENP-M, might take up the function of CENP-M in those organisms in which CENP-M cannot be identified.

Modeling of CENP-I suggests that it adopts an α-solenoid fold analogous to that observed in β-importin (*Cingolani et al., 1999*; *Vetter et al., 1999*). An N-terminal domain of CENP-I (residues 57–281) is sufficient to bind the CENP-H/K sub-complex, while the C-terminal half (residues 282-C) is necessary to bind CENP-M. Contiguity between CENP-H, CENP-I, and CENP-K had been previously

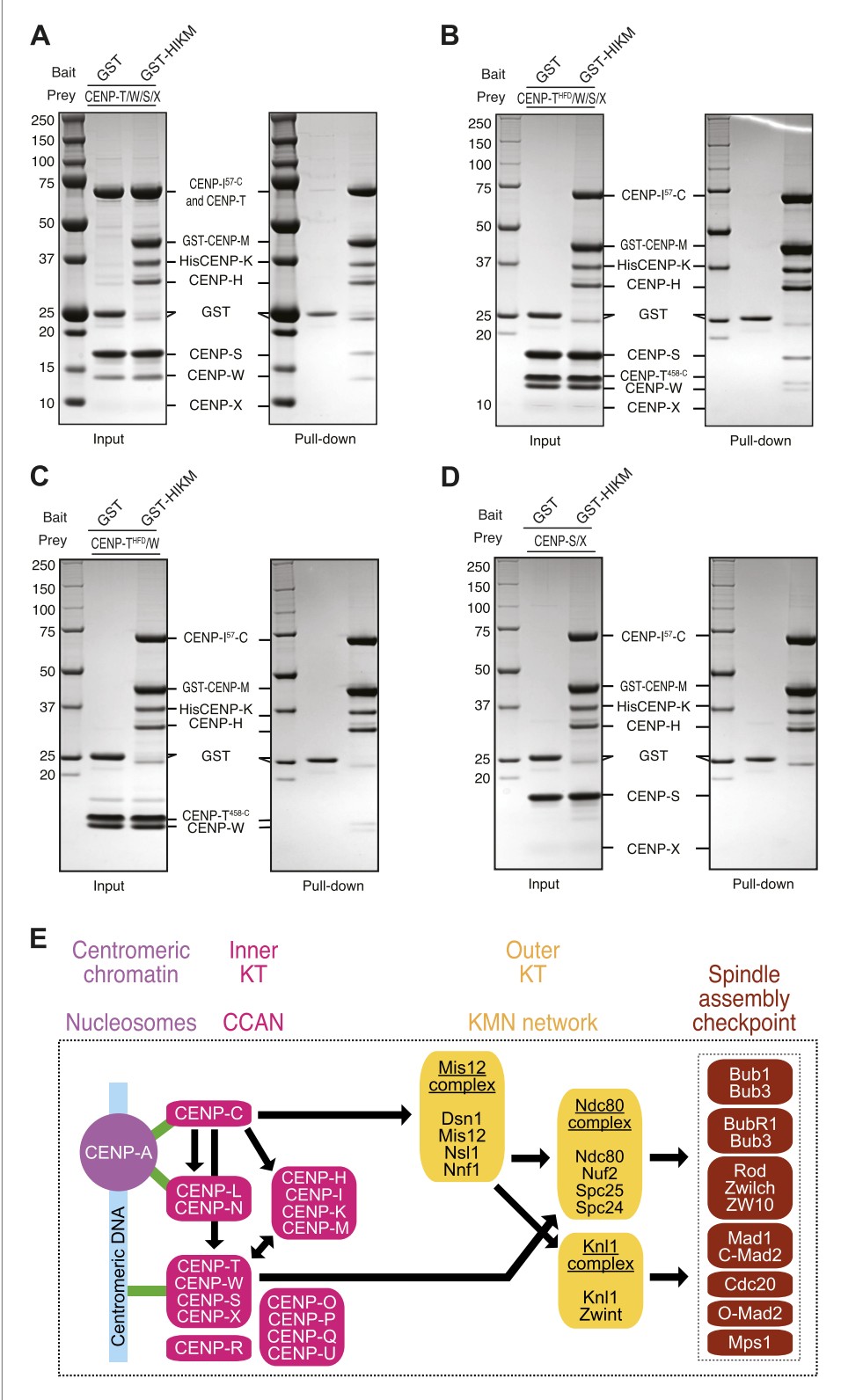

**Figure 7**. Direct interaction of HIKM complex with the CENP-T/W complex. (**A–D**) GST or GST-HIKM baits were immobilized on beads and incubated with (**A**) CENP-T/W/S/X complex, (**B**) CENP-T$^{452–C}$/W/S/X complex, (**C**) CENP-T$^{452–C}$/W, or CENP-S/X. For each sample, both the input and the solid phase bound material (indicated as

*Figure 7. Continued on next page*

*Figure 7. Continued*

'pull-down') are shown after separation by SDS-PAGE and staining with Coomassie brilliant blue. Note that full-length CENP-T and CENP-I[57–C] migrated indistinguishably. (**E**) Model of kinetochore assembly supported by our analysis. CENP-C and possibly CENP-N/L interact directly with the CENP-A nucleosome. The presence of CENP-C at the centromere is essential for the recruitment of CENP-T/W and CENP-HIKM complex. CENP-T/W and CENP-HIKM complex are co-dependent and interact physically with each other.

hypothesized based on proteomic analysis of precipitates from cellular lysates, from the similarity of phenotypes caused by depletion of individual subunits, and from 2-hybrid interaction data (*Measday et al., 2002*; *Okada et al., 2006*). However, that the interactions among these subunits were direct, and that CENP-M was also part of the complex, had remained unclear. Thus, our reconstitution of a stable quaternary complex, the HIKM complex, significantly extends these previous analyses and identifies a new stable sub-complex of crucial importance for kinetochore stability.

The HIKM complex flanks other previously recognized stable kinetochore sub-complexes, including the CENP-O/P/Q/U complex, the Mis12 complex, the Ndc80 complex, the CENP-L/N complex and the CENP-T/W/S/X complex (*Meraldi et al., 2006*; *Hemmerich et al., 2008*; *Perpelescu and Fukagawa, 2011*; *Westermann and Schleiffer, 2013*; *Westhorpe and Straight, 2013*). In this study, we have identified a direct interaction with the CENP-T/W/S/X complex. Our future studies will address whether HIKM forms direct interactions with other kinetochore sub-complexes or subunits. Also of interest is the role of the HIKM complex in recruiting the chromatin remodeling FACT complex, contributing to CENP-A deposition (*Obuse et al., 2004*; *Foltz et al., 2006*; *Okada et al., 2009*), as well as in the recruitment of spindle checkpoint components (*Liu et al., 2003*, *2006*; *Matson et al., 2012*).

CENP-T depletion hampers kinetochore localization of the subunits of the HIKM complex (*Foltz et al., 2006*; *Hori et al., 2008a*; *Gascoigne et al., 2011*). However, we now show that the HIKM complex is mutually required for stable association of CENP-T/W with kinetochores. The identification of a direct interaction between the HIKM complex and the CENP-T/W complex may configure a mechanistic basis for this phenomenon. Thus, the sole interaction with centromeric chromatin ('Introduction') is not sufficient to recruit or retain CENP-T/W at kinetochores in the absence of the subunits of the HIKM complex. Collectively, our observations, together with (a) recent structural data on DNA-bound CENP-T/W/S/X (*Takeuchi et al., 2014*), (b) previous observations that CENP-T and CENP-W undergo relatively rapid turnover times at centromeres (*Prendergast et al., 2011*), and (c) previous observations that CENP-S/X occupies a rather peripheral position within kinetochores (*Amano et al., 2009*) suggest that the hypothesis that CENP-T/W/S/X are embedded in a nucleosome-like structure (*Nishino et al., 2012*) might require further scrutiny.

Because CENP-C is required for recruitment of the HIKM complex (*Milks et al., 2009*; *Carroll et al., 2010*; *Gascoigne et al., 2011*), and HIKM is in turn required for CENP-T recruitment (this study), we tested and confirmed the prediction that CENP-T recruitment is also dependent on CENP-C. Conversely, disruption of the CENP-M/CENP-I interaction, or depletion of CENP-T, did not have major effects on the kinetochore localization of CENP-C, as previously suggested (*Goshima et al., 2003*; *Liu et al., 2006*). These findings unequivocally position CENP-T/W downstream of CENP-C, as previously proposed (*Carroll et al., 2010*), and imply that the interaction of CENP-C with CENP-A, possibly together with the interaction of CENP-L/N with CENP-A, represents the apex of the CCAN recruitment pathway (*Figure 7E*). Our future studies will aim to identify the molecular basis for this plan of kinetochore assembly.

## Materials and methods

### Expression and purification of CENP-M

A cDNA segment encoding human CENP-M isoform 1 was subcloned in pGEX-6P-2rbs, a modified pGEX-6P vector (GE Healthcare, Piscataway, NJ), as a 3' fusion to the sequence encoding GST. The construct CENP-M[1–171] was created by insertion of a stop codon with the QuikChange kit (Agilent Technologies, Inc., Santa Clara, CA). Constructs were sequence verified. The expression and purification procedure was the same for both CENP-M constructs. *Escherichia coli* C41 (DE3) cells harbouring vectors expressing CENP-M or CENP-M[1–171] were grown in Terrific Broth at 37°C to an $OD_{600}$ of 0.6–0.8,

when 0.2 mM IPTG was added and the culture was grown at 18°C for ~15 hr. Cell pellets were resuspended in lysis buffer (50 mM Tris/HCl pH 7.4, 300 mM NaCl, 5% glycerol, 1 mM DTT) supplemented with protease inhibitor cocktail (Serva, Heidelberg, Germany), lysed by sonication and cleared by centrifugation at 48,000×*g* at 4°C for 1 hr. The cleared lysate was applied to Glutathione Sepharose 4 Fast Flow beads (GE Healthcare) pre-equilibrated in lysis buffer, was incubated at 4°C for 2 hr, washed with 70 vol of lysis buffer and subjected to an overnight cleavage reaction with 3C protease to separate CENP-M from GST. Resource S cation exchange column (GE Healthcare) was pre-equilibrated in 20 mM MES pH 6.0, 50 mM NaCl, 5% glycerol, 1 mM DTT. The eluate from Glutathione beads was adjusted to a final salt concentration of 50 mM, loaded onto the Resource S column and eluted with a linear gradient of 50–500 mM NaCl in 10 bed column volumes. Fractions containing CENP-M were concentrated and loaded onto a Superdex75 SEC column (GE Healthcare) pre-equilibrated in SEC buffer (10 mM MES pH 6.0, 150 mM NaCl, 1 mM TCEP). Fractions containing CENP-M were concentrated, flash-frozen in liquid nitrogen and stored at −80°C.

## CENP-M$^{1–171}$ crystallization and structure determination

CENP-M$^{1–171}$ (10 mg/ml) was crystallized by sitting drop vapor diffusion using a Honeybee Cartesian robot and 96-well plates. Diffraction-quality crystals were obtained by optimizing the initial conditions in hanging drops. The optimal reservoir buffer contained 100 mM bicine pH 8.5, 11% MPD and 8 mM spermidine. Crystals were transferred to a cryobuffer containing the reservoir liquor supplemented with 15% glycerol and flash-frozen in liquid nitrogen. Selenomethionine (SeMet) derivatives were crystallized under similar conditions. X-ray diffraction data were collected with synchrotron radiation at beamline ID14-4 at the European Synchrotron Radiation Facility (ESRF, Grenoble, France) for the native crystal, and beamline X06DA (PXIII), Swiss Light Source (Villigen, Switzerland) for the SeMet crystal. X-ray diffraction data were processed with xia2 (version 0.3.3.1) (*Winter et al., 2013*). Analysis of data quality and crystal defects was performed using phenix.xtriage (*Adams et al., 2010*). Although the CENP-M crystals suffer from merohedral twinning, with a twinning fraction close to 50%, SAD phases obtained using Phenix AutoSol yielded an interpretable 2 Å experimental map. Model building was carried out in Coot (*Emsley et al., 2010*), with the help of fragments built automatically by Phenix AutoBuild, ARP/wARP (*Morris et al., 2004*) and Buccaneer (*Cowtan, 2006*). The model was then used for molecular replacement into the native dataset using Phenix AutoMR. Iterative model building with Coot and refinement with phenix.refine yielded a final model with two molecules covering the full asymmetric unit. The Collaborative Computational Project 4 (CCP4) suite (*Collaborative Computational Project, Number 4, 1994*) was also used at several stages. The structure was illustrated with PyMOL (www.pymol.org).

## Expression and purification of CENP-H/CENP-K complex

A cDNA segment encoding human CENP-K was subcloned in a MultiBac pFL-derived vector (*Fitzgerald et al., 2006*), with an N-terminal TEV cleavable 6xHis tag, under the control of the polh promoter. A cDNA segment encoding human CENP-H was subcloned in pUCDM vector, without any tag, under the control of the p10 promoter. Constructs were sequence verified. The two vectors were then fused via *in vitro* Cre-loxP recombination. Baculovirus was then produced as described previously (*Trowitzsch et al., 2010*), and amplified with three rounds of amplification. Expression of CENP-H/K complex was carried out in Tnao38 cells, using a virus: culture ratio of 1: 50 at 27°C for 72 hr. Cell pellets were resuspended in lysis buffer (50 mM Tris/HCl pH 8.0, 300 mM NaCl, 20 mM imidazole, 5% glycerol, 2 mM β-mercaptoethanol) supplemented with protease inhibitor cocktail (Serva), lysed by sonication and cleared by centrifugation at 48,000×*g* at 4°C for 1 hr. The cleared lysate was applied to Ni-NTA Agarose beads (Qiagen, Venlo, The Netherlands) pre-equilibrated in lysis buffer, was incubated at 4°C for 2 hr and washed with 70 vol of lysis buffer. Bound proteins were eluted with lysis buffer supplemented with 200 mM imidazole and then dialysed against 50 mM Tris/HCl pH 8.0, 150 mM NaCl, 5% glycerol, 0.5 mM EDTA, 1 mM DTT at 4°C overnight. During this dialysis step, tag cleavage with TEV protease was also performed. Resource Q anion exchange chromatography column (GE Healthcare) was pre-equilibrated in 50 mM Tris/HCl pH 8.0, 75 mM NaCl, 5% glycerol, 0.5 mM EDTA, 1 mM DTT. The dialysed sample was adjusted to a salt concentration of 75 mM, loaded onto the Resource Q column and eluted with a linear gradient of 75–500 mM NaCl in 10 bed column volumes. Fractions containing CENP-H/K complex were concentrated and loaded onto a Superdex200 SEC column (GE Healthcare) pre-equilibrated in SEC buffer (10 mM HEPES pH 7.5, 150 mM NaCl, 1 mM TCEP). Fractions containing CENP-H/K complex were concentrated, flash-frozen in liquid nitrogen and stored at −80°C.

## Expression and purification of CENP-HI[57–C]KM complex

Codon optimised human CENP-I 57-756 was subcloned in a MultiBac pFL-derived vector (*Fitzgerald et al., 2006*), with an N-terminal TEV cleavable 6xHis tag, under the control of the polh promoter. A cDNA segment encoding human CENP-M isoform 1 was subcloned in the 2nd MCS of the same vector, under the control of the p10 promoter. Simultaneously, a second pFL-based vector was created with untagged CENP-H and CENP-K under the control of the polh and p10 promoters, respectively. The CENP-I/M vector was then linearized with BstZ171, and the expression region corresponding to CENP-H/K was PCR amplified with primers designed for sequence and ligation independent cloning (SLIC) of the PCR fragment into the linearized CENP-I/M vector. The SLIC reaction was then carried out to produce a single pFL-based vector with four expression cassettes. Constructs were sequence verified. Baculovirus was then produced as described previously (*Trowitzsch et al., 2010*), and amplified with three rounds of amplification.

Expression of CENP-HI[57–C]KM complex was carried out in TnAo38 cells, using a virus:culture ratio of 1:40. Infected cells were incubated for 72 hr at 27°C. Cell pellets were harvested, washed in 1xPBS, and finally resuspended in a buffer containing 50 mM HEPES 7.5, 300 mM NaCl, 1 mM $MgCl_2$, 10% glycerol, 5 mM imidazole, 2 mM BME, 0.1 mM AEBSF, and 2.5 units/ml Benzonase (Millipore, Billerica, MA). Cells were lysed by sonication, and cleared for 1 hr at 100,000$g$. Cleared cell lysate was then run over a 5-ml Talon superflow column (Clontech, part of Takara Bio group, Shiga, Japan) and then washed with 50 mM HEPES 7.5, 1 M NaCl, 10% glycerol, 5 mM imidazole 2 mM BME. CENP-HIKM complex was eluted with a gradient of 5–300 mM imidazole, and the fractions containing HIKM pooled, and the His tag cleaved overnight at 4°C. HIKM in solution was then adjusted to a salt concentration of 100 mM and a pH of 6.5, prior to loading on a 6-ml Resource S ion-exchange column (GE Healthcare), equilibrated in 20 mM MES 6.5, 100 mM NaCl, 2 mM BME. CENP-HIKM was then eluted with a gradient of 100–1000 mM NaCl over 20 column volumes, and peak fractions corresponding to CENP-HIKM were pooled and concentrated in a 50 kDa MW Amicon concentrator (Millipore). CENP-HIKM was then loaded onto a Superdex 200 16/600 (GE healthcare) in 20 mM HEPES 7.5, 150 mM NaCl, 2.5% glycerol, 2 mM TCEP. Sample was concentrated and flash frozen in liquid $N_2$ prior to use.

## GST-CENP-M (wt and mutants), CENP-I[57–C], CENP-H and His-CENP-K co-expression in insect cells and GST-pull-downs

A cDNA segment encoding human CENP-M isoform 1 was subcloned in a MultiBac pFL-derived vector, with an N-terminal TEV cleavable GST tag, under the control of the polh promoter. Codon optimised human CENP-I 57-756 was subcloned in the 2nd MCS of the same vector, under the control of the p10 promoter. Mutant CENP-M constructs were created by site-directed mutagenesis using the QuikChange kit (Stratagene, La Jolla, CA). Constructs were sequence verified. Baculovirus was then produced and amplified with three rounds of amplification. The baculovirus encoding CENP-H/His-CENP-K complex, which has been detailed in the previous paragraph, was also employed. For each GST-pull-down experiment, 25 ml of freshly diluted Tnao38 cells at a density of $1 \times 10^6$ cells/ml in serum-free medium (Sf-900 II SFM; Life Technologies, Carlsbad, CA) were co-infected with GST-CENP-M/CENP-I[57–C] and CENP-H/His-CENP-K viruses using a virus: culture ratio of 1: 10 for each virus at 27°C for 72 hr. Cell pellets were resuspended in lysis buffer (20 mM HEPES pH 7.5, 300 mM NaCl, 1 mM TCEP) supplemented with protease inhibitor cocktail (Serva), lysed by sonication and cleared by centrifugation at 20,000×$g$ at 4°C for 30 min. The cleared lysate was applied to Glutathione Sepharose 4 Fast Flow beads (GE Healthcare) pre-equilibrated in lysis buffer, was incubated at 4°C for 2 hr, washed with 60 vol of lysis buffer and eluted with lysis buffer supplemented with 30 mM reduced Glutathione. Samples of total lysate, supernatant, beads before elution and elution were analysed by SDS-PAGE and Coomassie blue staining and by western blotting. The following antibodies were used: anti-CENP-M (in house made rabbit polyclonal antibody SI0868, raised against the full length protein; 1:1000), anti-CENP-I (rabbit polyclonal, ab28844; 1:100; Abcam), anti-CENP-H (goat polyclonal, sc-11297; 1:200; Santa Cruz, Dallas, Texas), anti penta-His (mouse monoclonal; 1:2000; Qiagen).

## In vitro protein binding to adenine and guanine nucleotides

N-methylanthraniloyl (MANT)-labeled nucleotides (ADP, ATP, GDP, GTP) (*Hiratsuka, 1983*), were purchased from Pharma Waldhof (Düsseldorf, Germany). The fluorescence quantum yield of MANT nucleotides increases in nonpolar solvents and upon binding to proteins. Fluorescence data were recorded with a Fluoromax-4 spectrophotometer (Jobin Yvon, Horiba, Kyoto, Japan), with excitation and emission

wavelengths of MANT-nucleotides at 366 and 450 nm, respectively. Arl2, a member of the Ras super-family of small GTPases, was used as control (kind gift of Mandy Miertzschke, Max Planck Institute of Dortmund, Germany). 500 µl of 1.0 µM MANT-labeled nucleotides in CENP-M SEC buffer were used. After 7 min, when the fluorescence baseline signal was stabilized, recombinant purified CENP-M or Arl2 (at 10 µM) was added and the fluorescence signal was monitored for 1 hr. For each experiment, the fluorescence signal was normalized to the fluorescence signal at time zero.

## Analytical SEC migration shift assays

Analytical SEC experiments were performed on calibrated Superdex200 5/150 or Superose6 5/150 columns (GE Healthcare). All samples were eluted under isocratic conditions at 4°C in SEC buffer at a flow rate of 0.2 ml/min for Superdex200 5/150 or 0.1 ml/min for Superose6 5/150. Elution of proteins was monitored at 280 nm. 100 µl fractions were collected and analysed by SDS-PAGE and Coomassie blue staining. To detect the formation of a complex, proteins were mixed at the indicated concentrations in 50 µl, incubated for at least 2 hr on ice and then subjected to SEC. For binding assays with nucleosomes, a SEC buffer containing 10 mM HEPES pH 7.5, 50 mM NaCl, 1 mM TCEP was used. For the other binding assays, a SEC buffer containing 150 mM NaCl was used when possible (namely, with CENP-M, CENP-H/K, His-CENP-I$^{57-281}$, CENP-H/K/I$^{57-281}$, Mis12 complex, Ndc80 complex, Knl1$^{2000-2311}$, Zwint). A SEC buffer containing 300 mM NaCl was instead employed with proteins that were not stable in lower NaCl concentrations (specifically, CENP-C constructs, CENP-T/W/S/X, CENP-L/N, CENP-O/P/Q/U, CENP-R).

## Cell culture and transfection

HeLa cells were grown in Dulbecco's Modified Eagle's Medium (DMEM; PAN Biotech) at 37°C in the presence of 5% CO$_2$ and supplemented with 10% Fetal Bovine Serum (FBS; Clontech), penicillin and streptomycin (GIBCO, Carlsbad, CA). Parental Flp-In T-REx HeLa cells used to generate stable doxycycline-inducible cell lines were a gift from Stephen Taylor (University of Manchester, Manchester, England, UK). Flp-In T-REx HeLa cells expressing CENP-M fusions to GFP were generated as previously described (*Tighe, 2004*) and maintained in DMEM with 10% tetracycline-free FBS supplemented with 250 µg/ml hygromycin and 4 µg/ml blasticidin (Invitrogen, Carlsbad, CA). GFP-CENP-M fusions were expressed by addition of 1 ng/ml or 50 ng/ml doxycycline (Sigma, St. Louis, MO) for 24 or 48 hr. For CENP-M silencing, we used a combination of three siRNA duplexes (target sequences: 5′-ACAAAAGGUCUGUGGCUAA-3′; 5′-UUAAGCAGCUGGCGUGUUA-3′; 5′-GUGCUGACUCCAUAAACAU-3′; Thermo Scientific, Carlsbad, CA) targeting the 3′-UTR of endogenous CENP-M. CENP-M siRNA duplexes were used at 20 nM each. For CENP-T silencing, we used a combination of two siRNA duplexes (target sequences: 5′-GUGGAGAAGUGCCUAGAUA-3′ from AMBION, and 5′-AAGUAGAGCCCUUACACGA-3′ from Thermo Scientific) at a concentration of 5 nM each. For CENP-C silencing, we used a single siRNA (target sequence: 5′-GGAUCAUCUCAGAAUAGAA-3′ from AMBION, Austin, TX) at a concentration of 7.5 nM. All transfections were performed with HyPerFect (Qiagen) according to the manufacturer's instructions. Phenotypes were analysed 66 hr (for CENP-T and CENP-C depletions), and 72 or 96 hr (for CENP-M depletions) after siRNA addition and protein depletion was monitored by western blotting or immunofluorescence. Where indicated, nocodazole (Sigma-Aldrich) was used at 0.3 µM for 16 hr, RO-3306 (Calbiochem, part of EMD Biosciences, Darmstadt, Germany) was used at 9 µM for 18 hr and MG-132 (Calbiochem) at 5 µM for 3 hr.

## Mammalian plasmids

A cDNA segment encoding human CENP-M isoform 1 was subcloned in pcDNA5/FRT/TO-EGFP-IRES vector, a modified version of pcDNA5/FRT/TO vector (Invitrogen, Carlsbad, CA) generated in house (*Petrovic et al., 2010*), as a C-terminal fusion to EGFP. Mutant CENP-M constructs were created by site-directed mutagenesis using the QuikChange kit (Stratagene). CENP-M cDNA was also subcloned in pcDNA5/FRT/TO vector (Invitrogen) as an N-terminal fusion to EGFP. Constructs were sequence verified.

## Immunoprecipitation and immunoblotting

To enrich cultures for mitotic cells, nocodazole was added to the cell culture media. Mitotic cells were then harvested by shake off and lysed by incubation in lysis buffer (75 mM HEPES pH 7.5, 150 mM KCl, 1.5 mM EGTA, 1.5 mM MgCl$_2$, 10% glycerol, 0.075% NP-40, 90 U/ml benzonase [Sigma], protease inhibitor cocktail [Serva] and PhosSTOP phosphatase inhibitors [Roche, Basel, Switzerland]) at 4°C for 15 min followed by sonication and centrifugation. Extracts were pre-cleared with a mixture of protein A-Sepharose (CL-4B; GE Healthcare) and protein G-Sepharose (rec-Protein G-Sepharose 4B;

Invitrogen) at 4°C for 1 hr. Subsequently, extracts were incubated with GFP-Traps (ChromoTek, Martinsried, Germany) at 4°C for 2–4 hr. Immunoprecipitates were washed with lysis buffer, resuspended in Laemmli sample buffer, boiled and analyzed by western blotting using 4–12% or 4–20% gradient gels (Life technologies). The following antibodies were used: anti-GFP (in house made rabbit polyclonal antibody; 1:4000), anti-Hec1 (mouse monoclonal, clone 9G3.23; 1:1000; Gene-Tex, Irvine, CA), anti-Mis12 (clone QA21; 1:1000; [*Petrovic et al., 2014*]), anti-Knl1 (in house made rabbit polyclonal antibody SI0787, raised against amino acids 1-22; 1:1000), anti-Vinculin (mouse monoclonal, clone hVIN-1; 1:15000; Sigma-Aldrich), anti-CENP-M (in house made rabbit polyclonal antibody SI0868, raised against the full length protein; 1:500), anti-CENP-I (in house made rabbit polyclonal antibody SI0887, raised against amino acids 57–281; 1:500), anti-CENP-T/W (in house made rabbit polyclonal antibody SI0882, raised against the full length protein complex; 1:800), anti-CENP-C (rabbit polyclonal antibody SI410; 1:1200; [*Trazzi et al., 2009*]). Secondary antibodies were affinity purified anti-mouse (Amersham, part of GE Healthcare), anti-rabbit (Amersham), anti-goat (Santa Cruz) conjugated to horseradish peroxidase (1:10000). After incubation with ECL western blotting system (GE Healthcare), images were acquired with ChemiBIS 3.2 (DNR Bio-Imaging Systems, Jerusalem, Israel). Levels were adjusted with ImageJ and Photoshop and images were cropped accordingly.

## Immunofluorescence and quantification

HeLa cells and Flp-In T-REx HeLa cells were grown on coverslips pre-coated with 15 μg/ml poly-D-Lysine (Millipore) and 0.01% poly-L-Lysine (Sigma), respectively. Cells were either fixed with methanol and rehydrated with PBS or fixed with PBS/PHEM-paraformaldehyde 4% followed by permeabilisation with PBS/PHEM-Triton 0.3%. The following antibodies were used for immunostaining: anti-Nsl1 [clone QM9-13; 1:1000], anti-Hec1 (mouse monoclonal, clone 9G3.23; 1:1000; Gene-Tex), anti-Knl1 (in house made rabbit polyclonal antibody SI0787, raised against amino acids 1-22; 1:1000), anti-CENP-M cross-linked to Alexa568 (in house made affinity purified rabbit polyclonal antibody SI0868, raised against the full length protein; 1:200), anti-CENP-I (1:700; a kind gift from Song-Tao Liu, University of Toledo, Ohio, USA), anti-CENP-T/W (in house made rabbit polyclonal antibody SI0882, raised against the full length protein complex; 1:800), CREST/anti-centromere antibodies (1:100; Antibodies Inc., Davis, CA), anti-Tubulin (mouse, 1:8000; T9026; Sigma). For CENP-C, either polyclonal anti-CENP-C (in house made rabbit antibody SI410; 1:1200) or monoclonal anti-CENP-C (mouse monoclonal, 2159C5a; 1:200; Abcam) were used. Cy3-conjugated, RhodamineRed-X-conjugated, Cy5-conjugated, and DyLigth649-conjugated secondary antibodies were purchased from Jackson ImmunoResearch Laboratories, West Grove, PA. Alexa 488-labeled and 568-labeled secondary antibodies were from Invitrogen. DNA was stained with 0.5 μg/ml DAPI (Serva) and coverslips mounted with Mowiol mounting media (Calbiochem) or ProLong Gold Antifade reagent (Life Technologies). All experiments were imaged at room temperature and, with the exception of *Figure 6F*, using the spinning disk confocal microscopy of a 3i Marianas system (Intelligent Imaging Innovations, Denver, CO) equipped with an Axio Observer Z1 microscope (Zeiss, Oberkochen, Germany), a CSU-X1 confocal scanner unit (Yokogawa Electric Corporation, Tokyo, Japan), Plan-Apochromat 63x or 100x/1.4NA objectives (Zeiss) and Orca Flash 4.0 sCMOS Camera (Hamamatsu, Hamamatsu City, Japan). Data for *Figure 6F* were acquired using a confocal microscope (model TCS SP2; Leica) equipped with a 63x NA 1.4 objective lens. Images were acquired as 0.27-μm Z-sections (using Slidebook Software 5.5 from Intelligent Imaging Innovations or using LCS 3D software from Leica, Solms, Germany) and converted into maximal intensity projections TIFF files for illustrative purposes. Quantification of kinetochore signals was performed on unmodified Z-series images using Imaris 7.3.4 software (Bitplane, Zurich, Switzerland). After background subtraction, all signals were normalized to CREST and values obtained for control cells were set to 1. Quantifications are based on two or three independent experiments where a minimum of 10 cells and 300 kinetochores per condition were analyzed.

## Electron microscopy and image analysis

Prior to EM experiments, the CENP-HI$^{57-C}$KM complex was separated on a Superdex 200 10/300 SEC column (GE Healthcare) (pre-equilibrated in 20 mM Tris/HCl pH 8.0, 150 mM NaCl, 1 mM TCEP) and diluted. 4 μl of the sample were adsorbed at 25°C for 40 s onto glow-discharged carbon-coated grids. The grids were washed twice with SEC buffer and negatively stained with 0.07% uranyl formate (SPI supplies/Structure probe, West Chester, PA) for about 120 s as described previously (*Bröcker et al., 2012*). Samples were imaged with a JEOL1400 microscope equipped with a LaB6 cathode operated

at 120 kV. Images were recorded at low-dose conditions (19 electrons/$Å^2$) at a corrected magnification of 82553x on a 4k × 4k CMOS camera F416 (TVIPS, Oslo, Norway). Single particles were manually selected, aligned, and classified using reference-free alignment and k-means classification procedures as well as the iterative stable alignment and clustering approach (ISAC) (*Yang et al., 2012*) implemented in SPARX (*Hohn et al., 2007*) and EMAN2 (*Ludtke, 2010*). The data set used for the analysis of the HIKM complex contained 5859 particles.

For the 3D reconstruction of the CENP-HIKM complex an initial 3D model was calculated from the best ISAC class averages using the SHC approach implemented in SPARX (*Hohn et al., 2007*). Then, using the complete set of 5859 raw particle images, the reconstruction was further refined by the iterative projection matching approach implemented in SPARX until convergence was achieved. The resolution of the final reconstruction was estimated by the 0.5 FSC criterion to be 22 Å (*Figure 4—figure supplement 3*, panel C). Chimera (*Pettersen et al., 2004*) was used for visualization, analysis, and preparation of EM figures.

## Sequence searches

To search for CENP-M homologous sequences, we first conducted PSI-blast searches using HsCENP-M. This identified a distant CENP-M ortholog from *Crassostrea gigas* (K1QTP9_CRAGI), whose sequence was then used for further iterative searches of the NCBI non-redundant database using hidden Markov models (HMM) (*Eddy, 2011*). At the third iteration, four sequences gave a significant match above threshold: a predicted protein from the sponge *Amphimedon queenslandica* (gi|340373098, e-value 0.00094), the GTP binding protein SAS1 from the fungus *Lodderomyces elongisporus* (gi|146447565, e-value 0.0036), a hypothetical protein from the amoeba *Dictyostelum purpureum* (gi|330794657, e-value 0.0017), and a Ras-related protein from the arthropod *Aedes aegypti* (gi|157117562, e-value 0.0092). We removed the redundancy of this alignment (50%) to keep the most representative sequences from the profile. Only common regions to the retrieved and CENP-M proteins present in the alignment were extracted (~140 amino acids). We then removed the four retrieved sequences from the alignment, built a new profile consisting only of 18 CENP-M sequences and searched the non-redundant database using hmmsearch (*Eddy, 2011*). The discarded sequences were again identified above threshold. Reciprocally, we then removed the CENP-M sequences from the alignment keeping only the four distant orthologs and generated a new HMM profile that was used to search the non-redundant database. The CENP-M protein from *Crassostrea gigas* (gi|405975648, or K1QTP9_CRAGI) was recovered at e-value 0.00034. Additional Blast (delta) searches with this protein provided CENP-M proteins. Given the large divergence in sequence similarity and the inability to obtain reliable sequence alignments for the N-terminal part of the CENP-M sequences with *bona fide* small GTPases, we created a sequence alignment from structural alignments. Rab1A (PDB 4FMC_D) was the best template found using CE algorithm (*Shindyalov and Bourne, 1998*) (3.6 Å root mean square deviation). The region spanning residues 66–156 of HsCENP-M displays the highest structural conservation between GTPases and CENP-M and identified reciprocal sequence similarities in iterative HMM searches. We derived a sequence alignment from the CENP-M/Rab1A structural alignment and built a profile from it. CENP-M, Rab1A, the identified GTPases, additional hits, and a selected group of sequences from the five classical GTPase families were aligned to the profile using hmmalign (*Eddy, 2011*). The alignment was visually inspected to detect inaccuracies and was used as input for phylogenetic inference using Maximum Likelihood (*Guindon et al., 2009*). PhyML was run using mpi implementation, with SPR and 5 starting trees. All parameters were optimized and 1002 bootstrap replicates were obtained from the tree (commands: mpirun -n 6 phyml-mpi -i 175toPhyML.phylip -b 1002 -d aa -m LG -f m -s SPR --rand_start --n_rand_starts 5 -v e -a e -o tlr). The tree was visualized using iTOL (*Letunic and Bork, 2007*) and is depicted in *Figure 3D*.

## Cross-linking analysis

CENP-HI[57–C]KM complex was cross-linked with isotope-labeled disuccinimidyl suberate and digested with Lys-C and trypsin after quenching with ammonium bicarbonate. Cross-linked peptides were enriched using size-exclusion chromatography, analyzed by liquid chromatography coupled to tandem mass spectrometry and identified by the search algorithm, xQuest. Cross-linking, MS analysis and database searching were performed as described (*Herzog et al., 2012*).

Visualization of the crosslinks was done by converting the raw data (in form of Excel spreadsheets) to the GEXF data format (Graph Exchange XML Format) using custom shell scripts. The data were

then imported into the Gephi software (http://gephi.org) that was modified to allow simultaneous calculation and display of curved and straight connectors (i.e., intra- and intermolecular crosslinks). The Gephi graph was exported as an Adobe Illustrator file for final processing.

### Data submissions
The final model of CENP-M has been submitted to the Protein Data Bank under the accession number 4P0T.

## Acknowledgements
We thank the staff of beamline X06DA (PXIII) and in particular Meitian Wang at the Swiss Light Source (PSI) and the staff of the European Synchtoton Radiation Facility (ESRF) for precious help in X-ray diffraction data collection, Fabrizio Martino and Daniela Rhodes for help with the reconstitution of recombinant nucleosomes, Mandy Miertzschke for help with GTP-binding assays, Alex Faesen for help with microtubule-binding assays, Song-Tao Liu and Stephen S Taylor for sharing reagents, Valentina Cecatiello for crystallization experiments, Giuseppe Ossololengo for production of polyclonal antibodies, Radovan Dvorsky for help with CENP-I modeling, Stephan Diekmann, Roger Goody, Gerben Vader, and the members of the Musacchio laboratory for helpful discussions. AM acknowledges funding by the European Union's 7th Framework Program ERC agreement KINCON and the Integrated Project MitoSys. FH is supported by the Bavarian Research Center of Molecular Biosystems and by a LMU excellent junior grant.

## Additional information

### Funding

| Funder | Grant reference number | Author |
| --- | --- | --- |
| European Research Council | KINCON | Andrea Musacchio |
| European Commission | Framework Program VII, MitoSys | Andrea Musacchio |
| Bavarian Research Centre of Molecular Biosystems | | Franz Herzog |

The funders had no role in study design, data collection and interpretation, or the decision to submit the work for publication.

### Author contributions
FB, SM, JRW, Conception and design, Acquisition of data, Analysis and interpretation of data, Drafting or revising the article; DP, AMR, TZ, ADA, BV, SG, LM, Acquisition of data, Analysis and interpretation of data; SJ, Analysis and interpretation of data, Drafting or revising the article, Contributed unpublished essential data or reagents; VK, Acquisition of data, Drafting or revising the article; AV, Analysis and interpretation of data, Drafting or revising the article; IRV, FH, SR, SP, Acquisition of data, Analysis and interpretation of data, Drafting or revising the article; AM, Conception and design, Analysis and interpretation of data, Drafting or revising the article

### Author ORCIDs
Federica Basilico, http://orcid.org/0000-0002-7002-6316

## Additional files

### Major datasets
The following dataset was generated:

| Author(s) | Year | Dataset title | Dataset ID and/or URL | Database, license, and accessibility information |
| --- | --- | --- | --- | --- |
| Basilico F, Pasqualato S, Weir J, Vetter I, Musacchio A | 2014 | Crystal structure of human centromere protein M | http://www.pdb.org/pdb/explore/explore.do?structureId=4p0t | Publicly available at RCSB Protein Data Bank. |

The following previously published dataset was used:

| Author(s) | Year | Dataset title | Dataset ID and/or URL | Database, license, and accessibility information |
|---|---|---|---|---|
| Dong N, Zhu Y, Lu Q, Hu L, Zheng Y, Shao F | 2012 | EspG-Rab1 complex | http://www.pdb.org/pdb/explore/explore.do?structureId=4fmc | Publicly available at RCSB Protein Data Bank. |

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
