## [Decision Letter]

Thank you for sending your work entitled “The pseudo GTPase CENP-M drives human kinetochore assembly” for consideration at *eLife*. Your article has been favorably evaluated by John Kuriyan (Senior editor) and 2 reviewers. The Senior editor and the two reviewers discussed their comments before we reached this decision, and the Senior editor has assembled the following comments to help you prepare a revised submission.

As you will see, the reviews are very positive and we will be pleased to move towards acceptance once the following issue is addressed. The principal issue has to do with the modeling of the CENP-I structure. While it seems reasonable, there is some concern that it is not properly validated and could lead to its acceptance as correct by the less than careful reader. Please read the reviews, which are included in their entirety below, and respond to us by email as to your planned course of action. The editor will discuss your response with the reviewers and will then give you a final set of instructions to submit a revised manuscript, which will be handled without further external review.

*Reviewer 1*:

The centromere and kinetochore are required for attaching chromosomes to the mitotic spindle, and thus for faithful chromosome segregation. Despite the importance of the centromere and kinetochore, how the >100 proteins of the complex come together to give rise to its essential functions is unknown. In this manuscript, Basilico and colleagues define the structure, function and interactions of an essential protein complex at centromeres the protein the CENP-HIKM complex. This paper is exceptionally well done and represents an important contribution to our understanding of human centromere structure and formation. It should be published with only minor modifications.

Basilico et al. first present a careful and thorough analysis of CENP-M, a poorly understood member of the centromere. The authors show that CENP-M and CENP-I interact and mutually depend on each other for centromere localisation. CENP-M is shown to be a member of a four-component complex also consisting of CENP-H/I/K, a previously suggested centromere subcomplex. What follows is an impressive biochemical, structural and bioinformatic characterisation of CENP-HIKM (HIKM). Indeed, the depth of analysis into HIKM separates this study from many previous studies of centromere components that often leave a confusing picture of the centromere.

In addition, the manuscript characterizes how HIKM 'fits' into the overall assembly of the centromere. While some aspects of this problem have been previously studied, the authors emphasize the observation that CENP-T/W/S/X requires HIKM for its centromere localization, and explain this with novel biochemical analyses that reveal HIKM directly interacts with CENP-T/W. This is an important missing link in our knowledge of centromere formation, and allows the authors to question some recent proposals that have confused the field.

In summary, the high quality of the data, the depth of the characterization of CENP-M/H/I/K, and the clarity of the presentation made this manuscript a pleasure to read. The literature on centromere formation presents a somewhat confusing and inconsistent picture, often due to a lack high quality biochemical analyses to complement cell biological approaches. This manuscript does not fall into this category. I highly recommend publication in *eLife*, with no significant changes or additions required.

Minor comments:

The authors use point mutations in GFP-tagged CENP-M to suggest that 'CENP-M and CENP-I are mutually required for kinetochore localization', as mutant CENP-M does not localize to kinetochores or rescue CENP-I localization (Figure 5). This model would be further supported by depleting CENP-I and showing complete loss of CENP-M from kinetochores. For example, if CENP-M had other roles at centromeres that didn't involve CENP-I, but did involve the CENP-M's L94 and L163 residues, I would expect the same results as those presented in Figure 5.

I am intrigued whether CENP-M RNAi has an effect on CENP-H & CENP-K? This would be an extension of Figure 4, which shows CENP-M RNAi prevents CENP-I localization. If HIKM was exclusively acting as a complex, an effect may be expected. If H/K were insensitive to CENP-M RNAi this would still be an interesting result.

The authors support a model that CENP-A nucleosomes (and their recognition by CENP-C/N) provides the central platform for centromere assembly (rather than the CENP-T/W/S/X complex). The primary data supporting this argument is the RNAi-mediated depletion of CENP-C resulting in CENP-T centromere delocalisation. This result was previously published and should be better referenced (7). However, the importance of this observation in the current manuscript helps to clarify a point of confusion in the field surrounding the interdependence of CENP-C and CENP-T/W/S/X and their roles in centromere assembly.

*Reviewer 2*:

Basilico et al present a comprehensive characterisation of the CENP-M protein, a constituent of the inner kinetochore and so-called CCAN network. The principle conclusions are that the fold of the protein is that of a degenerate GTPase and is required for the formation of a larger “HIKM” complex that acts upstream of several other CCAN components, notably the TWSX histone fold complex. These conclusions are supported by a variety of structural, biochemical and cell-based studies. In addition, a model for the HIKM complex based upon bioinformatic, cross-linking and EM studies is proposed. The data are well presented, thorough and of high quality. The notion that that the HIKM complex is upstream of TWSX recruitment and does not necessarily constitute a parallel pathway is also consistent with the evolutionary distribution of CCAN components. Overall, I think the study is eminently suitable for publication in *eLife*.

My principle concern with the manuscript is the modelling of the CENP-I structure, which is mainly based on secondary structure prediction algorithms. While it may prove to be broadly accurate, a more detailed bioinformatic analysis would strengthen the paper. For example, given that the predicted structure contains multiple HEAT repeats, can these then be identified on the basis of sequence alone, and aligned to each other? Are the predicted structures of Ctf3 and Mis6 similar to that of CENP-I (and do they contain putative repeats)? Presumably if the interaction with CENP-M occurs on the concave inner surface by analogy to the Ran-Importin structure, there should be conserved features present. Can these been identified and mapped to the modelled structure? I'm also not entirely convinced that the cross-linking data supports the N- to C- termini interaction as claimed. While there are indeed intramolecular cross-links found between these regions, there appear to be an almost equal number between the extreme N-terminal of the protein and the region around residue 300, implicated CENP-H/K binding. It is a little difficult to reconcile these observations with the model presented. Furthermore the schematic presented in Figure 4 implies that the coiled-coil of CENP-HK runs parallel to the outside surface of CENP-I (which is convex in the model) and would place the axis of the coiled-coil at right angles to the axis of the HEAT repeat, which seems unlikely (and also would probably not fit into the EM reconstruction as suggested in supplemental Figure 4). I appreciate that in the absence of experimental data, these types of modelling study are useful, but there is a danger of them becoming accepted as reality, particularly if determining the high-resolution structure of the complex proves intractable.

---

## [Author Response]

Reviewer 1:

*The authors use point mutations in GFP-tagged CENP-M to suggest that 'CENP-M and CENP-I are mutually required for kinetochore localization', as mutant CENP-M does not localize to kinetochores or rescue CENP-I localization (*Figure 5*). This model would be further supported by depleting CENP-I and showing complete loss of CENP-M from kinetochores. For example, if CENP-M had other roles at centromeres that didn't involve CENP-I, but did involve the CENP-M's L94 and L163 residues, I would expect the same results as those presented in*
Figure 5.

We agree with the reviewer that this experiment would nicely complement our data. We have therefore tried to perform this experiment by using silencing oligonucleotides designed based on a previous report (Liu et all, NCB 2003). We first tested the protocol described in the paper and that consisted in one round of transfection and a depletion time of 72 hrs. The results were disappointing, as the cells showed a very modest decrease in CENP-I levels. We therefore performed another experiment, using a protocol similar to the one used for CENP-M in our manuscript. Unfortunately, the depletion levels remained low (approximately a 20-30% decrease). It appears therefore that this experiment might require significant more preparation and troubleshooting than we had anticipated, and ultimately opted to submit the revised manuscript without it. We note however that Obuse *et al.* (Nature Cell Biology 2006) reported loss of CENP-M in DT40 chicken cells knockout for CENP-I, in line with the prediction that these proteins may be co-dependent for kinetochore localization.

*I am intrigued whether CENP-M RNAi has an effect on CENP-H & CENP-K? This would be an extension of*
Figure 4*, which shows CENP-M RNAi prevents CENP-I localization. If HIKM was exclusively acting as a complex, an effect may be expected. If H/K were insensitive to CENP-M RNAi this would still be an interesting result*.

Regretfully we are not in a position to carry out this experiment at this time, due to lack of a suitable antibody for recognition of CENP-H/K at the kinetochore by immunofluorescence. In the absence of such an antibody, which we have started producing to support a follow-up to this story, we would have to proceed by creating new cell lines with labeled versions of CENP-H or CENP-K, which, with all possible technical difficulties associated with the task, would clearly make the revision a rather lengthy process. In view of the fact that the reviewer considered this experiment a minor addition to the story we are presenting, we decided not to include it.

*The authors support a model that CENP-A nucleosomes (and their recognition by CENP-C/N) provides the central platform for centromere assembly (rather than the CENP-T/W/S/X complex). The primary data supporting this argument is the RNAi-mediated depletion of CENP-C resulting in CENP-T centromere delocalisation. This result was previously published and should be better referenced (*[7]*). However, the importance of this observation in the current manuscript helps to clarify a point of confusion in the field surrounding the interdependence of CENP-C and CENP-T/W/S/X and their roles in centromere assembly*.

We agree with the reviewer and in fact we had cited the work of [7] in the original version of the manuscript. We have now indicated in two additional instances the agreement of our results with the results of Carroll and colleagues (2010). In particular, we have included: “These results are in agreement with a previous report (7) and highlight the essential role of CENP-C as the basis of the pathway of kinetochore assembly of the HIKM and CENP-T/W complexes.” We then reinforce this as follows: “These findings unequivocally position CENP-T/W downstream of CENP-C, as previously proposed (7), and imply that the interaction of CENP-C with CENP-A, possibly together with the interaction of CENP-L/N with CENP-A, represents the apex of the CCAN recruitment pathway (Figure 7).”

Reviewer 2:

*My principle concern with the manuscript is the modelling of the CENP-I structure, which is mainly based on secondary structure prediction algorithms. While it may prove to be broadly accurate, a more detailed bioinformatic analysis would strengthen the paper. For example, given that the predicted structure contains multiple HEAT repeats, can these then be identified on the basis of sequence alone*, *and aligned to each other?*

We now present the output of the RADAR server in the context of Figure 4—figure supplement 1. The result of this search agrees with the possibility that CENP-I contains helical repeat elements. However, we cannot say for sure that CENP-I contains HEAT repeats or other helical repeats known to form helical solenoids. We argue that this is not surprising given the remarkable divergence of such repeats.

Are the predicted structures of Ctf3 and Mis6 similar to that of CENP-I (and do they contain putative repeats)?

We carried out additional I-TASSER modeling with the sequences of chicken CENP-I, Ctf3, and Mis6. The results, displayed in Figure 4—figure supplement 1, are indeed completely in line with the previous prediction based on the human sequence. In all three additional cases the searches converged on Importins as preferred initial models.

*Presumably if the interaction with CENP-M occurs on the concave inner surface by analogy to the Ran-Importin structure, there should be conserved features present*. *Can these been identified and mapped to the modelled structure?*

We report the results of this interesting analysis in Figure 4—figure supplement 2 and show that indeed the surface of CENP-I predicted to interact with CENP-M is highly conserved.

*I'm also not entirely convinced that the cross-linking data supports the N- to C- termini interaction as claimed. While there are indeed intramolecular cross-links found between these regions, there appear to be an almost equal number between the extreme N-terminal of the protein and the region around residue 300, implicated CENP-H/K binding. It is a little difficult to reconcile these observations with the model presented. Furthermore the schematic presented in*
Figure 4
*implies that the coiled-coil of CENP-HK runs parallel to the outside surface of CENP-I (which is convex in the model) and would place the axis of the coiled-coil at right angles to the axis of the HEAT repeat, which seems unlikely (and also would probably not fit into the EM reconstruction as suggested in supplemental*
Figure 4*). I appreciate that in the absence of experimental data, these types of modelling study are useful, but there is a danger of them becoming accepted as reality, particularly if determining the high-resolution structure of the complex proves intractable*.

The answer to the reviewer’s comments comes in two parts.

1) We repeated the coiled-coil prediction on CENP-H and CENP-K with a different server (REPPER, at http://toolkit.tuebingen.mpg.de/repper). Contrarily to program COILS, which we had originally used for our analysis, REPPER did not predict the existence of coiled-coils in CENP-H and CENP-K. We therefore suspect that the latter are merely helical proteins. At a suggestion of our colleague Andrei Lupas (who conceived both programs), with whom we discussed this issue, we opted to rely on the REPPER output and modified Figure 2 as well as our original discussion on the possible interaction of the coiled-coil regions predicted by COILS. We now refer to these as the “central regions” of CENP-H and CENP-K and present them as possibly helical regions.

2) We agree with the reviewer that there is a wealth of crosslinks between the N- and C-terminal regions of CENP-I and the middle region of CENP-I. However, our argument was based on the fact that the same residue of CENP-M interacts with both the N- and C-terminal regions of CENP-I, suggesting that they are close in space in the complex. This, together with the extensive number of intra-CENP-I crosslinks linking the N- and C-termini, supports rather convincingly the possibility that the N- and C-termini of CENP-I are close in space.

When considered altogether, our observations seem in line with the scheme of Figure 4, which of course is only meant as a useful “recap” of the observations that we can derive from our biochemical and biophysical analysis.